# Titanium Implants and Local Drug Delivery Systems Become Mutual Promoters in Orthopedic Clinics

**DOI:** 10.3390/nano12010047

**Published:** 2021-12-24

**Authors:** Xiao Ma, Yun Gao, Duoyi Zhao, Weilin Zhang, Wei Zhao, Meng Wu, Yan Cui, Qin Li, Zhiyu Zhang, Chengbin Ma

**Affiliations:** The Fourth Affiliated Hospital of China Medical University, Shenyang 110032, China; 2020121414@cmu.edu.cn (X.M.); 77704645@cmu.edu.cn (Y.G.); dyzhao@cmu.edu.cn (D.Z.); zhangweilin@cmu.edu.cn (W.Z.); wzhao89@cmu.edu.cn (W.Z.); 2019120901@cmu.edu.cn (M.W.); ycui80@cmu.edu.cn (Y.C.); qli@cmu.edu.cn (Q.L.); zyzhang@cmu.edu.cn (Z.Z.)

**Keywords:** titanium implants, local drug delivery system, bone regeneration, drug effect, titanium processing technology

## Abstract

Titanium implants have always been regarded as one of the gold standard treatments for orthopedic applications, but they still face challenges such as pain, bacterial infections, insufficient osseointegration, immune rejection, and difficulty in personalizing treatment in the clinic. These challenges may lead to the patients having to undergo a painful second operation, along with increased economic burden, but the use of drugs is actively solving these problems. The use of systemic drug delivery systems through oral, intravenous, and intramuscular injection of various drugs with different pharmacological properties has effectively reduced the levels of inflammation, lowered the risk of endophytic bacterial infection, and regulated the progress of bone tumor cells, processing and regulating the balance of bone metabolism around the titanium implants. However, due to the limitations of systemic drug delivery systems—such as pharmacokinetics, and the characteristics of bone tissue in the event of different forms of trauma or disease—sometimes the expected effect cannot be achieved. Meanwhile, titanium implants loaded with drugs for local administration have gradually attracted the attention of many researchers. This article reviews the latest developments in local drug delivery systems in recent years, detailing how various types of drugs cooperate with titanium implants to enhance antibacterial, antitumor, and osseointegration effects. Additionally, we summarize the improved technology of titanium implants for drug loading and the control of drug release, along with molecular mechanisms of bone regeneration and vascularization. Finally, we lay out some future prospects in this field.

## 1. Introduction

Bone is an irreplaceable organ; with high strength, it performs many important functions of the human body, including exercise, maintaining posture, and protection. In addition, it also plays an important role in the production and storage of blood cells, the storage of fats, minerals, and growth factors, and the regulation of acid–base balance. Therefore, some diseases that seriously affect bone structure and function may cause fatal results—for example, severe bone defects caused by bone infection, primary or secondary bone tumors, etc. These patients suffer from treatment pain and high treatment costs, seriously reducing their quality of life [1]. Globally, there are millions of orthopedic surgeries to treat these diseases every year, such as joint replacement, insertion of implants for bone defect repair, and total knee or hip replacement [2]. In these surgical operations, the use of implants has played an important role, mainly including metal implants (such as titanium alloys, stainless steel, chromium, nickel, tantalum, etc.), ceramics, and polymer materials (such as PEEK). These implants play a pivotal role in the field of orthopedics [3,4]. Although each implant material has its own unique advantages, titanium alloy has become the most common metal implant due to its excellent biocompatibility, low elasticity, and corrosion resistance [5].

In the later stage of treatment with titanium implants, the titanium oxide layer begins to absorb ions, proteins, and polysaccharides, and then osteoblasts and other immune cells and inflammatory cells begin to migrate to the bone implant, leading to tight bone attachment. The titanium implant–bone host interface is created, meaning that the titanium allows bone attachment and results in bone anchoring. This process is called osseointegration, and it is a key factor in the success of implants in the treatment of these serious orthopedic diseases [6]. However, it cannot be ignored that infection, bacterial biofilm formation, differentiation of bone cells, the qualitative and quantitative lack of bone at the recipient site, surgical trauma from implant insertion, limitations of the titanium surface, and metabolic changes in the bone will eventually lead to the failure of osseointegration [7,8,9,10,11]. In summary, the factors that affect the success of titanium implants are numerous.

Although titanium implants provide more treatment possibilities for the above-mentioned serious issues, orthopedic surgeons still face other unavoidable problems in complex treatments, such as prolonged hospital stays, implantation failure caused by infection, secondary surgery for removal of the implant, and the inability to personally target bone tumors and bone infections. [12]. In order to solve these unavoidable problems and avoid the failure of the titanium implants, the combined use of drugs is one of the possibilities. After implantation, systemic medication for the patient’s condition is usually prescribed by orthopedic surgeons, such as antibiotics, analgesic and anti-inflammatory drugs, antitumor drugs, and bone-cell-growth drugs [13]. Such drugs have become a conventional treatment strategy, along with individualized treatment of many patients’ diseases after implantation—such as bone-growth drugs (simvastatin, etc.) and anti-bone-catabolism drugs (calcitonin, etc.)—to cooperate with titanium implants in the treatment of large bone defects caused by osteomyelitis [14]. Analgesic and anti-inflammatory drugs (aspirin, etc.) relieve the postoperative pain of patients and control inflammation to a certain extent [15,16]. At the same time, antitumor drugs (Adriamycin, etc.) are also used in patients with bone tumors after extensive bone resection to control tumor progression [17]. It is very important to understand whether these drugs have an impact on the successful implantation of titanium implants, and this has not yet been summarized by previous research. These drugs may have “good” or “bad” effects on the osseointegration process of titanium alloys; they may affect the formation of biofilms by affecting the antibacterial properties of titanium alloys [18], or by affecting the adhesion and differentiation ability of bone cells on the titanium implant [19], or by affecting the surface or structural properties of the titanium implant [20], thereby determining the implant’s characteristics, biological events on the surface of the implant, and the final therapeutic effect.

However, systemic drug delivery still has its limitations, because systemic drugs are difficult to spread when delivered to highly calcified bone tissue through the circulating blood [21]. It is also difficult to reach the therapeutic concentration when the blood supply changes, such as in response to trauma or a tumor [22].

At present, ~90% of clinical drugs are hydrophobic and insoluble in water; at the same time, due to the inactivation or removal of organs such as the gastrointestinal system, kidneys, liver, etc., only 1% of the systemically administered drug generally reaches the site of interest [23]. A higher dose of the drug is required at this site to achieve an optimized local concentration. In order to solve the problems of tissue toxicity, low solubility, poor selectivity, and unfavorable pharmacodynamic limitations of personalized therapeutic drugs [24], in the past few years, much research has been carried out to develop more effective local drug delivery systems to compensate for the whole-body disadvantages of systemic drug delivery systems. Local drug delivery systems combine pharmacology and metal materials technology to make the implant load the drug through its coating, nanotube structure, covalent grafting, etc., and then perform local drug delivery around the implant, in order to achieve better results.

Compared with systemic drug delivery systems, the advantage of local drug delivery systems on titanium implants is that they can provide a better drug concentration to the bone microenvironment (the loaded agents can be directly released around the bone tissue) [14], improving the biological utilization rate (drugs from inactivating organs and bypassing the gastrointestinal barrier, etc.) [25,26] and providing more personalized disease treatment (the type of drugs loaded depends on the disease) [27], which can be regulated by the titanium implant according to individual requirements (it has previously been reported that different metal processing technologies can control the release and storage of drugs, etc.) [28]. The exploration and development of local drug delivery systems on titanium implants has gradually become one of the focal points in the field of orthopedics; it is also the focus of orthopedic surgeons. In summary, the presence of the local drug delivery system not only compensates for or enhances the target therapeutic effect of the titanium implant, but at the same time, the surface modification of the titanium implant, the coating constructed from organic and inorganic materials, and the change in the spatial structure or other metal processing technologies will also affect the storage and release of drugs in the local system, leading to the achievement of better therapeutic effects. The coordination of the rate of osseointegration and the timing of the drug delivery makes it possible to obtain a better therapeutic effect; they complement and promote one another in the treatment of implants in the field of orthopedics. This review focuses on the new exploration of this concept in pharmacology and metal and materials technology in recent years, and explores the mutual promotion of different drug-loading methods and different modified titanium implant in the field of orthopedics, along with their specific roles in the treatment of orthopedic diseases. Figure 1 shows that the detailed description of loading drugs, loading methods and processing technology of titanium, in addition to the function after implantation. From the current orthopedic clinical perspective, this review provides the experience and technology of previous researchers for the future clinical treatment of orthopedics, and presents prospects for the development of drug delivery systems and titanium implants.

### Local Drug Delivery Systems and Titanium Implants

Although the application of systemic drug delivery systems in clinical practice is very extensive, their limitations should not be ignored. In order to solve the current challenges, scholars have begun to focus on local drug delivery systems on titanium implants. Through the processing of titanium implants and the innovation of nano-drug-loading technology, the drugs can be loaded on the titanium implants and directly released into the bone under the control of the metal processing technology, which provides better drug concentration to the bone microenvironment, improves the bioavailability, and solves the problem of personalized treatment by loading drugs with different curative effects. The local drug delivery systems enhance or complement the various therapeutic capabilities of titanium implants, while the various technologies used to modify the titanium implants, in turn, control the storage and release of drugs, promote one another, and achieve better treatment effects [29,30,31]. As for the use of titanium implants in orthopedic applications, from the current orthopedic clinical perspective, orthopedic surgeons usually give more empirical consideration to the following aspects: (1) In order to ensure the success of implantation, the most fundamental objective is to avoid bacterial infection of the implant and reduce foreign body reactions. (2) After successful implantation, the problem of better osseointegration efficiency in patients with fractures or bone defects needs to be solved. (3) After achieving successful bone ingrowth, the primary diseases that cause fractures or bone defects—such as bone tumors and osteomyelitis—must be solved. In the whole process of this treatment, the biocompatibility and the antibacterial, antitumor, and osseointegration capabilities of titanium implants are their most essential properties. This review focuses on the synergistic use of local drug delivery systems and titanium implants to solve clinical issues such as antibacterial, antitumor, and osseointegration effects in recent years (the issues related to biocompatibility are explained in Section 2).

## 2. Antibacterial Effects

Although orthopedic surgeons pay great attention to ensuring that surgical operations are aseptic, there is still the possibility of bacterial invasion. In orthopedic clinics, the bacterial invasion of titanium implants is usually due to the following causes: (1) In bacterial invasion caused by open trauma, for example, the bacteria remain in the epidermis, subcutaneous tissue, or deep tissue, and it is usually difficult to completely eliminate all bacteria. (2) The patients’ diseases, such as bacteremia, enable bacteria to adhere to the surface of the titanium implant via the circulatory blood. (3) The operation does not meet the necessary requirements, or the incision becomes infected after the operation, leading to the invasion of bacteria. (4) Bacterial invasion has already occurred during the preparation or transportation of the titanium implants. In short, once a bacterial invasion occurs, it will occur in the deep tissue around the titanium implant, making failure of implantation inevitable, and the effect of disinfection and systemic antibiotics will be minimal [32,33,34]. Staphylococcus aureus, Staphylococcus epidermidis, and Pseudomonas aeruginosa are the most common pathogens in titanium implant infections [35], and these bacteria are usually found in the titanium implants and accumulate, finally forming a hard biofilm (a matrix of hydrated polysaccharide secreted by the bacteria). The biofilm forms a microenvironment that supports the bacteria and protects them from host defense systems and antibacterial drugs. As the metabolic activity of biofilm-resident bacteria decreases, their sensitivity to most antimicrobial agents also decreases [36]. A structured layer of bacterial biofilm is shown in Figure 2A [37]. At the same time, because the bacteria firmly adhere to the surface of the titanium implant and compete with the bone cells for the surface of the implant, the bone cells cannot normally adhere to the surface of the implant, and the subsequent bone growth and osseointegration cannot be carried out, resulting in implantation failure [38]; therefore, the bacterial infection after titanium implantation can be fatal, and the revision surgery to fix the implantation will bring treatment pain and economic burden to the patient. This also further shows the necessity of using antibiotics after surgery; the use of antibiotics can stop the accumulation of bacteria and reduce the possibility of biofilm formation [32].

However, the systemic administration of antibiotics after surgery may cause the antibiotics to not accumulate at a high enough concentration around the titanium implant to achieve antibacterial and bactericidal effects; not only that, high concentrations of systemic antibiotics can also cause harm to other tissues of the body. However, local drug delivery systems on titanium implants allow higher concentrations of antibiotics to penetrate the biofilm and bone tissue, solving the aforementioned problems to a certain extent. Here, we mainly take vancomycin as an example to summarize the progress of titanium implants in coordinating local delivery of drugs while jointly improving the antibacterial properties in recent years. Vancomycin is a glycopeptide antibiotic; it is suitable for the treatment of serious and life-threatening Gram-positive bacterial infections, and has been widely used to treat and prevent osteomyelitis and deep infections; it is part of a key group in the structure of bacterial cell walls. Peptidoglycan interferes with the synthesis of cell walls, inhibits the synthesis of phospholipids and peptides in the cell wall, inhibits the growth and reproduction of bacteria, has no cross-resistance with other antibiotics, and has very few drug-resistant strains [39,40]; because of its excellent pharmacological properties, in recent years, it has been favored by researchers in topical drug delivery systems for titanium implants. As for the content we reviewed, it is precisely because of the antibacterial properties of titanium-based implants that their drug delivery systems are loaded with these antibiotics. Naturally, the loading of these antibiotics is completed before surgery. Due to the broad-spectrum antibacterial properties of vancomycin, the type of bacterial invasion after implantation is unknown. Vancomycin becomes the first choice when a decision must be made [41]. Interestingly, in the 2019–2021 articles in this field, most authors chose vancomycin for follow-up experiments; therefore, we chose vancomycin as a representative to conduct a review in this field, but gentamicin and first-generation cephalosporins have also been reported by scientific researchers.

### 2.1. Anodizing Titania Nanotubes and Vancomycin

Fathi et al. reported that they prepared a TiO_2_ nanotube layer on the surface of an implant via electrochemical anodization technology, and then loaded vancomycin into the TiO_2_ nanotubes and coated them with silk fibroin nanofibers; the scheme is shown in Figure 2B. They aimed to improve the titanium implants’ properties and the factors controlling drug delivery. On the other hand, by affecting the parameters of the electrospinning process, the size of the silk fibroin nanofibers can also be changed, thereby controlling the sustained and long-term release of vancomycin in the TiO_2_ nanotubes; a cross-sectional image of the SF nanofiber coating on the TiO_2_-NTs, along with the loaded drug vancomycin, is shown in Figure 2C. In the results of cell experiments and animal experiments, due to the high nanosurface roughness of the TiO_2_ nanotube structure, both showed a good effect in promoting the growth of bone cells and osseointegration. Due to the coating of silk fibroin nanofibers, the burst release of vancomycin in TiO_2_ nanotubes decreased from 81% to 29%, and release was sustained for 7 days [42]. The local sustained release of vancomycin can not only reduce the toxicity of antibiotics to other tissues, but also enable the local bone microenvironment to reach the required vancomycin concentration to control the bacterial cell wall synthesis and biofilm formation in order to achieve better antibacterial effects. However, the benign reactions are mutually propelled, reducing the competition between bacteria (and the biofilm formed) and bone cells on the surface of the titanium implant, which also has a positive effect on the osseointegration effect. There are also similarities with the reports of Liu et al. [43].

**Figure 2 nanomaterials-12-00047-f002:**
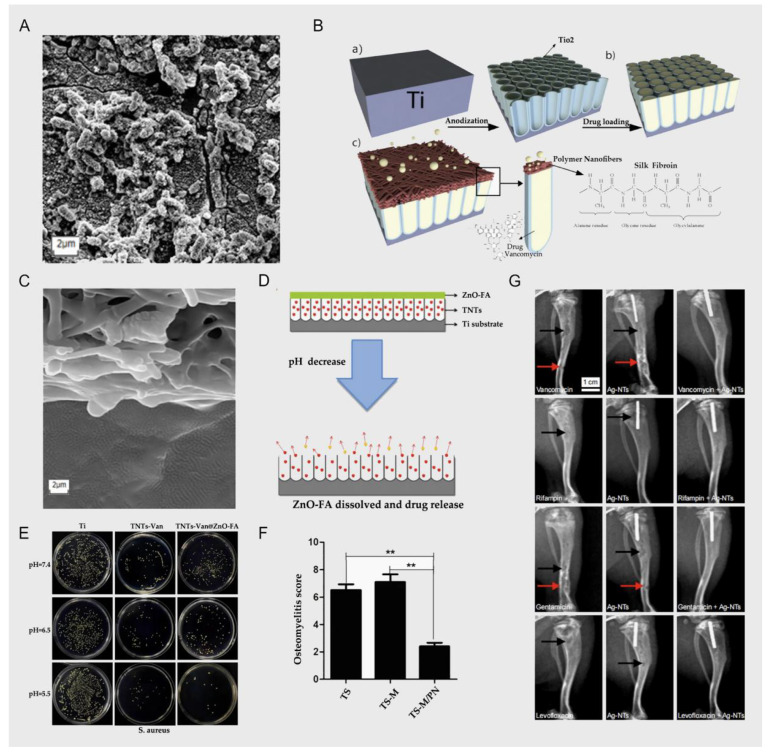
(**A**), *P*. *aeruginosa* biofilm on platinum; scale bar = 2 μm; reprinted with permission from [36]; copyright 2020 Kirchhoff et al. (**B**) Experimental scheme of Fathi et al. (**a**) Bare TiO_2_-NTs layer created on Ti substrate by electrochemically anodizing; (**b**) loading of Vancomycin within TiO_2_-NTs structures; (**c**) SF Nanofibers coated on TiO_2_-NTs to control Vancomycin release, antibacterial properties and enhance bone integration. The scheme presents a diffusion of Vancomycin molecules via SF Nanofibers; reprinted with permission from [42]; copyright 2019 Elsevier B.V. (**C**) A cross-sectional image of the SF nanofiber coating on TiO_2_-NTs (along with loaded vancomycin); reprinted with permission from [42]; copyright 2019 Elsevier B.V. (**D**) Experimental scheme of Xiang et al.; reprinted with permission from [44]; copyright 2018 Elsevier B.V. (**E**) Different antibacterial activity between the experimental group and the control group at different pH values; reprinted with permission from [44]; copyright 2018 Elsevier B.V. (**F**) The osteomyelitis scores of the experimental group and the control group (** denotes *p* ≤ 0.05); reprinted with permission from [29]; copyright 2020 The Royal Society of Chemistry. (**G**) Images showing X-ray examination 3 weeks post-surgery; the rats in group I (antibiotic) and group II (Ag-NTs) exhibited classic symptoms of implant infection, including bone absorption (black arrow) and fibrosis (red arrow); the group III (Ag-NTs + antibiotic) rats showed no signs of infection; reprinted with permission from [45]; copyright 2017 Dove Press Ltd.

Xiang et al. reported that the anodized TiO_2_ was loaded with vancomycin, and the tops of the TiO_2_ nanotubes were covered with some functionalized ZnO complexes; they combined folic acid and ZnO through an amidation reaction to produce the compound ZnO-FA, which would dissolve in the weak acid environment (after bacterial infection) and then free Zn^2+^; the scheme is presented in Figure 2D. Due to this design, if there is a bacterial infection around the implants, the pH will gradually reduce. The free Zn_2_^+^ and the burst release of vancomycin caused by the decomposition of ZnO-FA will synergistically kill the bacteria, and the degree of decomposition of ZnO-FA changes with the change in the pH value [44]. As shown in Figure 2E, there are significant differences in antibacterial activity between the experimental group and the control group at different pH values; the authors’ ingenious design integrates Zn^2+^ with bactericidal properties and a controlled release of vancomycin with a change in pH value (changing with the degree of bacterial infection); this design is an improvement in some respects compared with those of Fathi et al. and Liu et al. TiO_2_ nanotubes also have excellent biological properties as containers for vancomycin. The surface structure of TiO_2_ nanotubes also allows bone cells to adhere, differentiate, and proliferate more rapidly, thereby achieving a better osseointegration effect. The above technology can improve the release duration of vancomycin from a few hours to several days or tens of days, which not only prevents toxicity to the surrounding tissue caused by explosive release, but also—because of the long-term high concentration of vancomycin around the titanium implant—prevents the attachment and aggregation of microorganisms.

In the report of Xu et al., the authors also display similar thinking as in the reports above; they still used titanium nanotubes to store vancomycin, and prepared Ag-loaded nanoparticles on titanium implants so that Ag^+^ and vancomycin would release around the implant together. Because the Ag^+^ has bactericidal properties, it can cooperate with vancomycin to achieve a better antibacterial effect, as confirmed by the results of the animal experiments shown in Figure 2G, where the infection in the experimental group was milder and the osseointegration effect was better [45]. Compared with the reports of Fathi et al. and Xiang et al., Xu et al.’s design paid more attention to the early antibacterial properties of titanium implants. According to clinical experience, the risk of bacterial infection is indeed the greatest within 3 days after surgery [46]. The antibacterial performance of the implant can be further enhanced by other antibacterial or bactericidal agents in conjunction with vancomycin. This can be corroborated by the experiments of Croes et al. [47] and Aunon et al. [48].

In this field, the process of preparing the surface of titanium nanostructures via anodic oxidation technology is very common. TiO_2_ nanostructures have been proved to have a positive effect on the attachment and regeneration of bone cells [49]. The ingenious designs above are based on TiO_2_ nanotubes that were designed as a container for vancomycin. However, more effective drug release “switches” and precise regulation of drug release rates require reasonable drug loading and metal processing technology design, which will be the main direction of future research in this field.

### 2.2. Electrochemical Deposition and Vancomycin

Zhang et al. used 3D printing technology and micro-arc oxidation technology to successfully prepare titanium implants loaded with vancomycin. With a larger surface area (spatial high-porosity structure), the electrochemical technology of micro-arc oxidation was used to form an oxide ceramic coating on the surface of the implants for loading vancomycin. This technology increases the loading capacity of vancomycin to a certain extent; as the result of the process, the load and release are more stable; this scheme is shown in Figure 3A. The technique shows a good therapeutic effect in the rabbit tibial osteomyelitis model; in Figure 2F, we can see that compared with the control group, osteomyelitis was significantly controlled, and the osseointegration effect was significantly enhanced. From the perspective of physical modification of titanium implants, the innovative macroporous and high-porosity 3D-printed titanium implants meet the requirements of bone conduction and stability, but this is precisely because of the large surface area of their internal spatial structure, which provides a better space for the attachment and proliferation of bacteria, increasing susceptibility to bacterial infection [29]. However, the release of vancomycin leads to the inhibition of bacterial cell wall synthesis, prevents bacterial reproduction and the formation of biofilms, and significantly reduces the risk of bacterial infection. On the other hand, combined with porous titanium micro-arc oxidation technology, due to the loading of heparin and the polydopamine coating on the porous titanium surface, the implant can effectively store vancomycin and heparin molecules and continuously release them. To a certain extent, this achieves the PH-responsive release of vancomycin, making the release of vancomycin further controllable [29]. Similarly, in the report of Bezuidenhout et al., vancomycin was released from the channels of cementless titanium alloy cubes through a polyethersulfone membrane; the material’s structure is shown in Figure 3B, where the opening channels of the titanium cube with internal channels are fixed by a membrane. Although there are many differences in metal processing technology, their loading of vancomycin displays similar thinking in terms of release regulation. The larger spatial surface area of this high-porosity titanium implant scaffold is more conducive to enhancing the osseointegration effect, but this spatial structure also gives bacteria a superior environment, making bacterial infections more common. Therefore, in order to solve this problem, many scholars choose electrochemical techniques such as micro-arc oxidation to stably load antibiotics and other antibacterial agents on the surface of the implant in order to combat bacterial infections [50]. In the report of Li et al. [51], there were similar explorations. The use of electrochemical deposition technology to deposit vancomycin on the surface of titanium implants with a large spatial surface area is a breakthrough technology, but it still has its limitations. A series of chemical reactions may affect the efficacy of antibiotics, and due to the large spatial surface area of the implant, it is difficult to control the release. Follow-up research is still needed for improvement of this technology.

Furthermore, Chernozem et al. reported that while combining the aforementioned anodization and electrochemical deposition techniques, they paid more attention to improving the biocompatibility of titanium implants, which was also beneficial in improving the antibacterial properties of the titanium implants. In order to balance the various complex reactions after implantation and improve the biocompatibility, Chernozem et al. prepared an anodized TiO_2_-NT surface, and then used electrochemical deposition technology to deposit synthesized Ag NPs and CaP NPs on the TiO_2_-NT surface. Since the surface of TiO_2_-NTs is hydrophilic, the application of Ag NPs leads to a decrease in the water Ca and an increase in the surface free energy due to the increased contribution of the polar component, whereas the surface of biocomposites with CaP NPs is superhydrophilic. The characteristics of the above titanium implants lead to better biocompatibility and antibacterial properties, as has also been verified in subsequent cell experiments. At the same time, the authors demonstrated the fabrication of Ag and CaP NPs, which inhibit the growth of bacteria and can be used for the functionalization of titania NTs [52]. Chernozem et al. also emphasized the importance of biocompatibility on the basis of enhancing the antibacterial properties of implants. Generally speaking, biocompatibility refers to the degree of mutual acceptance of materials, living tissues, and bodily fluids—that is, the degree of foreign body reaction. The current research on the biocompatibility of titanium implant materials mainly focuses on the following three aspects: (1) the overall physiological impact of titanium implants on tissues and organs; (2) the metabolic process of the degradable part of titanium implants in the body; and (3) the effect of titanium implants on information transmission and gene regulation among cells, tissues, and organs [53,54,55]. The molecular composition and structure of the surface of the biometal material strongly affect the composition and structure of the protein it adsorbs, so their subtle changes can significantly change the biological activity of the material. The material can be modified via the surface modification of titanium implants and other processing techniques, effectively controlling the surface. However, the current research is mainly focused on (1) (as per the report of Chernozem et al.), while there are few studies on (2) and/or (3). The biocompatibility of titanium implants requires deeper exploration in the future [56,57].

It cannot be ignored that, in recent years, covalent grafting technology has gradually matured—usually by covalently linking some high-molecular-weight polymers and functionalized polymers to achieve antibacterial properties, such as polyNaSS polymers, hyaluronic acid, etc. [59]. Pichavant et al. reported that they used covalent grafting technology to successfully connect vancomycin to titanium implants, and in vivo experiments confirmed its superior antibacterial properties [60]. However, unlike the previous two reports, the titanium implant used by Pichavant et al. was a titanium plate instead of a titanium alloy scaffold. It may be the case that covalent grafting technology encounters technical difficulties in high-porosity structures (such as the complex internal spatial structure of a titanium implant). However, in Pichavant’s report, the loading and release of vancomycin had a better effect. How to apply covalent grafting technology to titanium alloy scaffolds with high porosity and complex spatial structure should be a direction for future research.

### 2.3. Chemical Coating and Vancomycin

Ordikhani et al. reported that they prepared a drug-eluting coating based on chitosan containing different amounts of vancomycin on titanium implants via a cathodic electrophoretic deposition process; Figure 3C shows the top-view and cross-sectional SEM images of the material they prepared. Under the process of electrochemical deposition and electrophoretic deposition, the titanium implant has nanometer-scale morphological characteristics and better wettability, which can be used to regulate the release of chitosan. Representative topographical AFM images of chitosan and the drug-eluting composite coating are presented in Figure 3D. In their vitro experiments, when the coating was loaded with 174 μg/cm^2^ of vancomycin, the number of colonies in the titanium implant was reduced by 85%, and the survival rate was lower at higher concentrations. However, considering the toxicity to the surrounding tissues, the authors believe that vancomycin at this concentration reaches a balance in many aspects. At the same time, they studied the release rate of vancomycin in the coating to draw their conclusions. They divided the release of vancomycin from the chitosan coating into three steps: The first step is the removal of the physical encapsulation of vancomycin in the hydrogel network, resulting in a burst of rapid release. The second step is the stable release under the influence of the number of chitosan coating layers. In the third step, the degradation and desorption of chitosan in the later stage causes the release of vancomycin to slow down, and the above results can be confirmed by the experimental results shown in Figure 3E [30]. In the reports of Swanson et al. [61], Ordikhani [62] et al., Rahnamaee [63], and Liu et al. [64], although different metal processes and drug-loading technologies were used, they all used chitosan or hyaluronic acid for vancomycin storage and as a medium for slow-release control. The three steps of vancomycin storage and release are similar to the above summary. The use of chitosan, hyaluronic acid, and other common loading materials on the surface of titanium implants for loading and controlled release of vancomycin (and other antibiotics) has not been uncommon in recent years. However, there remain problems in achieving the more effective control of the explosive release of the “first step”, and whether the duration of the stable release of the “second step” can be fully extended. This will depend on the progress of metal material processing technology and drug-carrying technology, and further studies are needed.

In recent years, layer-by-layer self-assembly technology has received more and more attention from scholars. Nancy et al. reported that their pure titanium was electrophoretically modified using double-layer coatings consisting of TiO_2_-SrHAP as the first layer followed by vancomycin-incorporated chitosan/gelatin as the second layer; the two layers are attracted to one another through electrostatic force, and each layer has functional groups with different properties, achieving a composite performance. As Nancy et al. concluded, one layer is used as a storage space for vancomycin, and the other layer is a “switch” that controls the release of drugs; a schematic representation of the step-by-step process involved in fabricating single- or double-layer coatings is shown in Figure 3F [58]. This technique was similarly used in the reports of Ionita et al. [65] and Zarghami et al. [66]. At present, the most important problem is how to control the stability of the mechanical properties of each film in the LbL technology. This is very important for clinical applications, and needs to be studied and resolved.

## 3. Antitumor Effects

The pathological fractures of some orthopedic patients are caused by primary or secondary bone tumors. These patients with bone tumors usually need implants to fix the fracture after massive bone tumor resection, while it is necessary to treat the bone tumors at the same time [67]. However, after surgery, systemic chemotherapeutics usually require higher doses to achieve effective concentrations around bone tumors. Because of that, patients also suffer from the toxicity of chemotherapy drugs. In addition, due to the changes in blood supply caused by surgical trauma and tumors, systemic administration is difficult to deliver to the bone tumors even at higher doses [68]. Scholars have proposed a solution. The implantation of titanium implants with anti-bone-tumor drugs has become a breakthrough choice, which not only enhances the possibility of bone regeneration after bone defects, but also protects patients from systemic toxic effects. Under these circumstances, the tumor is directly exposed to the chemotherapy drugs at the required concentrations. This section will introduce the progress of local drug delivery systems for titanium implants in collaboration with clinically common doxorubicin, curcumin, cisplatin, etc., to synergistically achieve the purpose of antitumor action.

### 3.1. Titanium Nano/Micro-Surface Modification and Antitumor Activity

Maher et al. reported that they used 3D printing and anodizing technologies to prepare a unique micron- and nano-scale titanium surface morphology—that is, a TiO_2_ nanotubes structure—and load doxorubicin (DOX) through the TiO_2_ nanotubes and the gaps between the tubes. Doxorubicin is a broad-spectrum antitumor antibiotic that can inhibit the synthesis of RNA and DNA, and is widely used in the treatment of osteosarcoma. In the report by Zhang et al., the authors used doxorubicin and cisplatin in combination to treat osteosarcoma, and obtained better cell morphological results [69,70]. At the same time, the combined use with paclitaxel also has a good antitumor effect, so there are still prospects to be explored for the better exertion of the antitumor ability of doxorubicin [71]. In particular, the results of adhesion and aggregation of fibroblasts, as shown in Figure 4A, are even more surprising. The accumulation of fibroblasts on the implant and the surrounding granulation tissue become the precursor of a bony callus, which gradually forms via the continuous deposition of calcium salt crystals in the implant, accelerating the repair of bone tissue around the implant [72]. The results of in vitro experiments showed that DOX loaded into TiO_2_ nanotubes was slowly released within 16 days, and maintained the required drug concentration in the tumor microenvironment; the drug release curve is shown in Figure 4B, indicating a significant inhibitory effect on osteosarcoma cells. In addition to antitumor activity, the nano/micro-scale surface morphology of the titanium implant enhances protein adsorption and osteoblast activity, thereby improving osteoblast adhesion and long-term osseointegration [31]. A similar technique was used in the report of Zhang et al., who successfully prepared curcumin-loaded functionalized titanium-based implants. Curcumin is a yellow pigment extracted from the rhizome of the ginger plant turmeric, which has been applied to the treatment of osteosarcoma by researchers, and induces mitochondrial dysfunction caused by excessive production of ROS in osteosarcoma cells which, in turn, leads to osteosarcoma cell apoptosis [73]. Zhang et al. loaded curcumin into titanium dioxide nanotubes modified with cyclodextrin polymer (pCD and TiO_2_ nanotubes are both used for curcumin storage), and a polydopamine coating was used as an auxiliary film to ensure the reliable anchoring between the cyclodextrin polymer and the TiO_2_ nanotubes; the scheme is shown in Figure 4C. This ingenious design allows curcumin to be slowly released within 90 h after implantation. In vivo experiments have also obtained better results in terms of anti-osteosarcoma effects, as confirmed in Figure 4E where, compared with the control group, the diameter of the tumor size in the experimental group was significantly reduced. Additionally, functionalized titanium implants with a surface density of 22.48 g·cm^–2^ support the attachment and proliferation of osteoblasts in vitro, which significantly improves the biocompatibility of implants [74]. In the study of Kaur et al., the microstructure of TiO_2_ nanotubes was also prepared via anodizing and sonoelectrochemical techniques, and the voltage control in anodizing was closely related to the morphology of the titanium nanosurface (i.e., the depth and diameter of the TiO_2_ nanotubes). After they loaded doxorubicin, they studied at which anodic oxidation voltage the doxorubicin had the best drug storage space and sustained release time in the TiO_2_ nanotubes. The results show that anodizing at 60–75 V has the best effect, and the antitumor ability is the strongest at this time [75]. According to the above research, In the field of TiO_2_ nanotubes loaded with antitumor drugs, more precise metal processing technology will bring more suitable drug-carrying containers, resulting in a better antitumor effect (the larger the spatial structure inside the titanium implant, the larger the drug load between the titanium nanotube and its gap, etc.). Moreover, better drug-coating technology can also assist the sustained, slow, and precise release of drugs (coatings, nano-drug loading, pH-, light-, and electricity-responsive inorganic compound deposition, etc.), which needs to be further explored.

### 3.2. Chemical Coating and Antitumor Activity

Jing et al. reported that they loaded the first-line clinical anti-osteosarcoma drug cisplatin into the PLGA-polyethylene glycol-PLGA temperature-sensitive hydrogel and used it to coat the titanium implant. Therefore, a bone substitute with anti-osteosarcoma and bone repair functions was constructed. Cisplatin is one of the most widely used antitumor drugs; its center is a heavy metal complex composed of divalent platinum, two chlorine atoms, and two ammonia molecules, making it similar to a bifunctional alkylating agent. Cisplatin can inhibit the process of DNA replication, and is usually given by intravenous injection to treat osteosarcoma. However, its intravenous administration has several problems, including nephrotoxicity, bone marrow suppression, nausea, vomiting, and drug concentrations that are difficult to reach in the tumor microenvironment. Clinically, these problems have been solved by intraperitoneal, intra-arterial, and intratumoral drug delivery, as well as implant-loaded local chemotherapy [78]. In the study of Jing et al., a PLGA-polyethylene glycol-PLGA temperature-sensitive hydrogel was used for the stable storage of cisplatin. The local temperature of the implant can be used as a “switch” for the release of cisplatin, as schematically represented in Figure 4D. The results of experiments in a rat tumor model show that the implants of the experimental group had better anti-osteosarcoma effects and fewer side effects, and also had a positive effect on osteogenesis [76]. However, the simple hydrogel covering the surface of the titanium implants may have the problem of unstable bonding and easily slipping off after being exposed to external force. Nevertheless, the chemical reaction of the strong bonding may still cause the properties of the hydrogel and cisplatin to change. These challenges require further research. A similar technology was used in the report of Sarkar et al., who used plasma spraying technology to cover the titanium implant with a hyaluronic acid coating that stores curcumin and vitamin K2, while the drug release of the hyaluronic acid layer changes with the change in the pH value; all schemes are shown in Figure 4F. Therefore, in the acidic microenvironment of osteosarcoma, the hyaluronic acid coating can continuously release curcumin and vitamin K2 into the osteosarcoma microenvironment due to its unique properties. From the results of in vitro experiments, the dual release of curcumin and vitamin K2 showed few or no osteosarcoma cells attached to the implant. Compared with the control group, the survival rate of osteosarcoma cells on the implants was reduced by 95% and 92% on the 7th day and the 11th day, respectively. At the same time, the drug synergy of curcumin and vitamin K2 also played a positive role in osteogenesis, as confirmed by the results shown in Figure 5A [77]. The application of plasma spraying technology leads to better sealing performance of the coating and the titanium implants, and enhances the wear and corrosion resistance of the material [79]. In the report of Zhang et al., the authors used induction suspension plasma spraying technology to prepare a hydrogenated black TiO_2_ (H-TiO_2_) coating with a micro/nano-scale morphology and an excellent and controllable photothermal effect, which plays a role in the treatment of tumors. The results of in vivo and in vitro experiments show that the H-TiO_2_ coating prepared under 808 nm near-infrared laser irradiation can inhibit tumor growth in vivo and in vitro, and can obtain a better antitumor effect. This means that H-TiO_2_ coatings may be a promising implant material for the treatment of bone tumors and bone regeneration, but the authors did not continue to combine this technology with local drug delivery systems. It can be inferred that combining H-TiO_2_ coatings with the sustained local release of antitumor drugs may bring better dual therapeutic effects [80], but it is possible that such a combination could bring unexpected results.

## 4. Osseointegration Effect

As a biologically inert material, the process of bone cell adhesion and ingrowth on the surface of the titanium implant is called osseointegration. Osseointegration begins with the absorption of ions, proteins, polysaccharides, and proteoglycans by the titanium oxide layer, and then macrophages, neutrophils, and osteoblasts (mainly osteoblasts) migrate to the bone–implant interface and cause bone adhesion in close contact with the implant surface [86]. In clinical practice, successful bone ingrowth and osseointegration are the basis for successful implantation—especially for patients with large bone defects caused by trauma, osteomyelitis, or tumor resection—requiring faster bone growth and better osseointegration in addition to conventional treatments that lead to higher requirements for implants [87]. The osseointegration effect of the titanium implant surface depends on the micro/nano-scale characteristics and chemical composition of the implant surface. Different implant surface characteristics and chemical compositions can affect the adsorption of proteins, and can stimulate the migration of osteoblasts and the adhesion of fibroblasts, which can affect the osseointegration effect [88]. In addition to the cytological mechanism (osteoblasts, osteoclasts, fibroblasts, macrophages, neutrophils, etc.), the histological mechanism cannot be ignored. Accompanied by the regeneration of bone cells, the endothelial cells around the implant are regulated by angiogenesis activators or pro-angiogenic factors (e.g., aFGF, bFGF, VEGF, etc.), and the vascular endothelial cells arranged along the blood vessels accelerate the proliferation, leading to the formation of new blood vessels [55,89]. Vascular invasion promotes the transport of nutrients, wastes, and precursor cells for growing/regenerating bone tissues; it also supports crosstalk between blood vessel endothelial cells and precursor cells to promote osteoblastic differentiation. Vascularization of peri-implant tissue is also very important to the remodeling and preservation of bone around an implant after placement [90,91,92,93,94,95,96]. Therefore, the regeneration of bone cells at the cytological level and the formation of blood vessels at the histological level are used to construct nutrient transport channels, which lead to better osseointegration effects. On the other hand, the use of drugs to promote osseointegration has gradually received more attention from scholars. Compared with the systemic administration of drugs to promote osseointegration before and after implantation, a local drug delivery system that carries osseointegration drugs (i.e., drugs promoting bone growth or angiogenesis) on titanium implants has a better effect. This section focuses on the local drug delivery of titanium implants in collaboration with the commonly used clinical drugs calcitriol, indomethacin, simvastatin, bisphosphonates, and VEGF (there are almost no commercialized drugs for promoting angiogenesis) to synergistically accelerate the progress of osseointegration, along with the loading and release technology of the aforementioned drugs.

### 4.1. Titanium Nano/Micro-Scale Surface Modification and Osseointegration

Kwon et al. reported that they used anodizing technology to prepare a spiral-shaped titanium implant with a chemically controlled titanium dioxide nanotube surface structure via immersion and drying under vacuum, as shown in Figure 5B, where the zoledronate is loaded inside the lumen of the titanium dioxide nanotube. Zoledronate is very common in clinical applications in orthopedics; it is a nitrogen-containing bisphosphonic acid compound that has a high affinity for mineralized bone. After systemic administration, it can selectively act on bones, target farnesyl pyrophosphate synthase in osteoclasts, and then directly induce osteoclast apoptosis by inhibiting the activity of osteoclasts, thereby inhibiting the resorption of bone cells. At the same time, zoledronic acid can also inhibit the increase in osteoclast activity and the release of bone calcium induced by a variety of stimulating factors produced by tumors, so it is usually used in clinical practice for lytic bone metastasis of malignant tumors [97]. The implants were taken out three weeks after implantation in the rabbits; compared with the control group, the titanium implants in the experimental group loaded with zoledronate had greater torsion resistance, and more new bone was formed around the implants. Therefore, it can be confirmed that zoledronate, as an anti-bone-catabolism drug, has the effect of accelerating osseointegration when locally administered via titanium implants. At the same time, due to the loading of the titanium nanotube structure, zoledronate can be slowly released locally from the implant for up to 3 weeks [81]. Similar conclusions were reached in the report of Sul et al., who also loaded zoledronate into fluorinated TiO_2_ nanotubes via surface modification technology. In animal experiments, it was also found that the loading of zoledronate enhanced the osseointegration effect; however, the authors suggested that the biochemical bond formed between fluorinated TiO_2_ and zoledronic acid may enhance the osseointegration effect. Since zoledronate is positively charged, the electrostatic interaction between zoledronic acid and fluorinated TiO_2_ nanotubes may also be one of the reasons for the enhanced osseointegration effect. This also proves that different nano-surface modification and chemical modification of titanium implants, in synergy with different osseointegration drugs, will have different effects [98].

He et al. reported that they used 3D printing technology to prepare a porous titanium scaffold modified with TiO_2_ nanotubes, and simulated the layered trabecular bone structure; the whole scheme is shown in Figure 5C. Anodizing technology was used to construct a drug-loading area and load calcitriol into the titanium nanotubes and the gaps between them, and then coat the thermosensitive Pluronic F-127 hydrogel to control the slow release of calcitriol. Calcitriol is the 1α,25-dihydroxy metabolite that exerts the strongest anti-rickets activity through the metabolism of vitamin D3 by hydroxylase in the liver and kidneys; it is absorbed by the small intestine, and can stimulate the activity of original osteoblasts or accelerate the formation of new osteoblasts, promote bone absorption, and transfer calcium and phosphorus into bone cells, thereby accelerating bone formation and osseointegration on the titanium implant [99]. The conclusion showed that compared with the control group, the slow release of calcitriol around the implant resulted in an increase in new bone around the implant and an enhanced osseointegration effect, confirming the positive effect of calcitriol in promoting osseointegration via the local drug delivery system. The structural characteristics of the 3D-printed porous titanium scaffold—similar to those of layered trabecular bone—also play a role in accelerating osseointegration. However, compared with the control implants, this faster osseointegration efficiency gradually loses its advantage two weeks after implantation. The authors believe that this may be related to the slow release of calcitriol in the titanium nanotubes for 14 days, which resulted in the lack of local delivery of calcitriol to promote bone anabolism, so the osseointegration slowed down after 2 weeks. The above conclusion can be confirmed by the results shown in Figure 5D. This limitation was proposed by the authors, and is one that they hope to resolve in follow-up research [82]. In the report of Gulati et al., a similar 3D printing technology was also used and optimized, with which they prepared a titanium alloy implant with a unique dual morphology (comprised of micron-sized spherical particles and vertically arranged TiO_2_ nanotubes); scanning electron micrograph (SEM) images of anodized 3D-printed Ti are shown in Figure 4E. The difference from the report of He et al. above was that the preparation of micron-sized spherical particles changed the surface morphology of the implant, leading to a significant increase in the calcitriol loading of the titanium implant compared to the surface morphology of pure TiO_2_ nanotubes. Since Gulati et al. did not prepare a coating that controls the sustained release of calcitriol, the specific conclusions on the sustained release capability cannot be compared, but this is also a breakthrough for the original technology [83].

### 4.2. Chemical Coatings on Titanium Surfaces and Osseointegration

Lai et al. reported that they used layer-by-layer deposition technology to prepare loaded chitosan, gelatin, and simvastatin on a titanium implant’s multilayered substrate coating. Simvastatin—a dual anabolic and anti-catabolic drug—is a HMG-CoA reductase inhibitor, and is widely used as a hypolipidemic drug; however, it also plays an important role in the regulation of bone metabolism. Simvastatin inhibits the formation of osteoclasts induced by RANKL and inhibits their apoptosis via the β-receptor mechanism [100]. In the multilayered coating structure, chitosan is used as a polycation layer, the gel is used as a polyanion layer, and simvastatin is loaded in the gap between the two layers. The electrostatic force makes the loading of simvastatin more stable. Chitosan and the gel layer are used to control the release of simvastatin through layer number control and other methods—that is, as the multilayer film is degraded layer by layer, simvastatin is released to the vicinity of the implant in a controlled manner to play a role. The results of in vitro experiments proved that the experimental group implants have great potential to inhibit the proliferation and differentiation of osteoclasts and promote the proliferation of mesenchymal stem cells. Moreover, the release of simvastatin is controlled by the multilayered coating structure, achieving a slow release that can last up to 14 days, meaning that the concentration of simvastatin around the endophytes is always maintained at an appropriate level; the release curve of simvastatin is shown in Figure 5F [84]. In response to this problem, a report by Stein et al. pointed out that in the absence of an optimal concentration, local delivery of simvastatin may induce an inflammatory response in titanium implants, which is detrimental to the osseointegration effect [101]. This conclusion indicates that when simvastatin is used in the local delivery systems of titanium implants to enhance the osseointegration effect, it is very important to control the release of simvastatin to ensure the appropriate concentration. However, although the multilayered coating structure used in the report by Lai et al. could control the sustained release of simvastatin, the control time was only 96 hours, which is far from the clinical requirements.

In order to ensure local, sustained, and long-term release of zoledronate by the implant, a new type of bone conduction implant was prepared. Mesoporous TiO_2_ has high surface area, large pore volume, and controlled meso-scale porosity. Because of the nature of mesoporous TiO_2_, when combined with anodizing technology, it can load more drugs and control their slow release. The authors prepared a mesoporous TiO_2_-layered titanium implant and loaded it with a large amount of zoledronate. The experimental results showed that zoledronate can be released locally in MLT for up to 21 days, and due to the control of the metal processing technology in the internal spatial structure of the MLT, different spatial structures also determine the release rate of the loaded zoledronate [102]. As an anti-bone-catabolism drug, zoledronate promotes osseointegration locally. This technology has made a contribution to the control of the long-term release of loaded drugs, but on the other hand, due to the larger surface area and greater porosity of the titanium implant, other properties—such as antibacterial properties—are reduced; however, these still need to be explored.

Peter et al. reported that they prepared titanium implants loaded with zoledronate with a hydroxyapatite coating. Hydroxyapatite is widely used as a titanium implant coating; it is used for the storage of zoledronate, released into the surrounding bone microenvironment after implantation, and the grafting technology on the coating can also be more stable and loaded with more zoledronate. According to the results of animal experiments, it was concluded that the increase in bone density around the titanium implant is dependent on the concentration of zoledronic acid. The release of zoledronic acid from the hydroxyapatite coating positively affects the structure of trabecular bone, thereby enhancing the osseointegration effect and the mechanical stability and strength of the implant [103]. In the report of Jakobsen et al., zoledronate was also used in the local drug delivery systems of titanium implants; they found that the local administration of zoledronate on poly(D,L-lactide) (PDLLA)-coated implants can also enhance the osseointegration effect of the implants, as confirmed by Figure 5G [85].

Leedy et al. reported that they designed a VEGF-loaded chitosan coating for titanium implants, and the chitosan was chemically bonded to the surface of the titanium implant by silane–glutaraldehyde. Vascular endothelial growth factor (VEGF)—a heparin-binding growth factor specific to vascular endothelial cells—is a powerful growth factor that promotes angiogenesis as well as osteoblast differentiation and bone regeneration [104]. The release of VEGF is controlled by chitosan, and the release duration is ~14 days, which is divided into a burst release time of 12 hours, a plateau release of 3 days, and a slow release of 11 days. The advantages of osseointegration and vascularization have been confirmed in cell experiments [105]. A similar design was also used in the report of Mullin et al. [106]. However, as a whole, there is less exploration of vascularization in this field, and this direction is worthy of follow-up research and exploration.

In summary, there is an issue that has to be taken into account—after implantation, the release of drugs under the control of implant preparation technology and the processes of bone aggregation and osseointegration on the surface will begin at the same time, and they are bound to affect one another. The release of drugs that promote bone regeneration and osseointegration will make bone aggregate, adhere, and regenerate on the surface of the titanium implant. The process of osseointegration is enhanced by the action of drugs [107]; at the same time, with the gradual formation of bone trabeculae, tissue vascularization, the increase in the number of bone cells that grow in the titanium alloy pores, and bone mineralization, the development of the bone microenvironment around the drug-loaded titanium implant is inevitable. Significant changes will have a certain impact on the process of drugs’ release, diffusion, and absorption into the blood [108]. Conversely, the above process can also cause the concentration change around the titanium implant to affect the healing of the bone. The relationship between the release of drugs under the control of implants and the process of osseointegration is extremely complicated. The objective is to balance the drug release and the osseointegration effect of the titanium implant. The current body of research in this field is relatively small, and this remains an issue to be explored in the future.

## 5. Discussion

From the clinical perspective of orthopedics, the development of local drug delivery systems for titanium implants has given orthopedic treatment a new, feasible, and effective direction, which inevitably must find a way to balance the following four challenges: the pharmacological properties of loaded drugs, the molecular biological mechanisms of bone cells and the surrounding tissues, the processing technology of titanium implants, and the correct choice of clinical application.

### 5.1. Loaded Drugs and Their Pharmacological Properties

In the clinical practice of orthopedics, the drugs frequently used with titanium implant surgery are mainly divided into the following categories: analgesic and anti-inflammatory drugs (NSAIDs, acetaminophen, and opioids are the most common), anti-bone-tumor drugs (e.g., doxorubicin, cisplatin, and curcumin), antibiotics (e.g., vancomycin and clindamycin), anabolic drugs (e.g., calcitriol, teriparatide, and fluvastatin), anti-catabolic drugs (e.g., alendronate sodium, zoledronate, and raloxifene), and dual anabolic and anti-catabolic drugs (e.g., simvastatin and strontium ranelate). Due to the pharmacological properties of these drugs, they are usually selected as systemic drugs or drugs loaded in titanium implants to achieve the purpose of personalized treatment. However, many reports have confirmed that the properties of these drugs sometimes affect titanium, including the osseointegration effect of the implant interface, which deserves attention. NASAIDs control inflammation and pain by regulating the COX-2 pathway in arachidonic acid, thereby reducing the synthesis of prostaglandins, but Chikazu et al. and Zhao et al. reported that the lack of COX-2 can cause limited adhesion and reduce the aggregation of osteoblasts around titanium implants in mice, thus weakening the effect of osseointegration [109,110]. However, in the report of Yang et al., it was further suggested that the controlled dose of NSAIDs has an effect on the osseointegration effect [111]. In response to the above problems, the advantages of acetaminophen are shown, as it has a low effect on the COX-1 and COX-2 pathways; although it is weak in inhibiting the synthesis of prostaglandins, it still does so to some extent, has and also exerts a certain anti-inflammatory and analgesic effect. This may make acetaminophen an appropriate choice for patients with non-severe pain who need to enhance the rate of osseointegration, which may be helpful for the subsequent choice of anti-inflammatory drugs for titanium implants. Teriparatide is a kind of anabolic drug that acts on the PTH receptor of osteoblasts, thereby affecting the level of parathyroid hormone to regulate bone production. However, when teriparatide is applied in large doses, it can cause extensive activation of osteoclasts, and will cause the expression levels of specific transcription factors, osteocalcin, bone salivary protein, and type I collagen in osteoblasts to be downregulated to varying degrees, tending toward bone catabolism. These conclusions are corroborated in the reports of Almagro et al. and Tao et al. [112,113]. Therefore, after the titanium implant is loaded with teriparatide, it is necessary to strictly control the local concentration of the drug around the implant. Zoledronate is an excellent anti-catabolic drug, which has a high affinity for mineralized bone; it targets farnesyl pyrophosphate synthase in osteoclasts, and then inhibits the activity of osteoclasts or directly induces their apoptosis in order to inhibit their bone resorption [114]; thus, scholars often choose it as a loading drug for titanium implants. However, according to reports by Basudan et al. and Cardemil et al., zoledronate has different osseointegration effects on bone tissues in different parts of the human body; for example, it has negative effects on the mandible, tibia, and fibula. Therefore, the application of zoledronic acid in different implant sites needs to be individually adjusted. The pharmacological properties of the drug itself and the titanium implants may interfere with one another. Today, there is relatively little exploration in this field; for example, it is unknown whether antitumor drugs, opioids, etc., will affect the properties of titanium implant, and, thus, affect osseointegration. It is necessary to be more cautious in the choice of implant loading drugs, and further experiments are needed. In addition, scholars have paid more and more attention to melatonin in the treatment of bone tumors in recent years, but the use of melatonin in the local drug delivery of titanium implants is rare [115].

### 5.2. Molecular Biological Mechanisms of Bone Cells and the Surrounding Tissues

From the perspective of molecular biology of osteoblasts, osteoclasts, fibroblasts, and mesenchymal stem cells, the mechanisms of bone aggregation, osteoinduction, and osseointegration on titanium implants have been deeply studied [116,117,118,119]. However, in recent years, there have been more and more explorations of the role of integrins on the surface of bone cells. Olivares-Navarrete et al. reported that they prepared titanium implants with a graphite carbon coating, and studied the important role of integrins in the process of bone cells’ maturation and osseointegration on the implants’ surface. Integrin is a transmembrane heterodimer, and each subunit has a unique role. It is because of the existence of integrin that osteoblasts and mesenchymal stem cells are connected with extracellular proteins, including actin, laminin, etc.; thus, integrin regulates the subsequent important processes of bone cell attachment, aggregation, and migration. The experimental results of this study also indicate that the integrin α subunit plays a major role in surface chemical recognition, and within a certain range, the roughness of the titanium implants can enhance the effects of bone differentiation and osseointegration [120]. The exploration from the perspective of molecular biology also provides instructions for the selection and application of implants. For example, due to the surface chemical recognition of integrin α subunits, it is possible to choose loading drugs or other agents to target the integrin α subunit so as to affect the maturation of osteoblasts. It is also possible to reasonably increase the surface roughness of the titanium implant in order to enhance the osseointegration effect. In order to play a prompting role, the molecular biological mechanisms of bone growth, osteoinduction, and osseointegration on the surface of titanium implants still need to be further explored. In addition, as for the related issues of vascularization at the histological level (relevant mechanisms are elaborated in Section 4.1), there are few studies and few related commercial drugs; this is expected to become a key research direction in the future.

### 5.3. Processing Technology of Titanium Implants

The physical modification of titanium implants, in terms of surface modification and other processing technologies, plays an important role in their antibacterial properties and their ability to promote osseointegration. It has been shown that previous researchers prepared TiO_2_ nanotube structures to carry drugs via different metal processing techniques—such as the micro/nano-level modification of the titanium surface—and adjusted the diameter, vertical depth, and tube wall thickness of the titanium nanotubes to control the drug load or its release, as well as via preparation of ceramic thin film layers by electrochemical deposition to store drugs, and by preparing a coating structure on the surface of the implant, using the ability of the coating structure to load the drug and control its release. Generally speaking, the common challenges focus on how to ensure the stability of drug loading and long-term slow drug release, and whether the physical properties of titanium implants—such as elastic modulus, corrosion resistance, and fatigue resistance—can meet the needs of patients. At the same time, under different conditions, the requirements for the pore size, porosity, and internal spatial structure of titanium implants (titanium scaffolds) are also different. Under this complex structure, the control of drug loading and release has become a new challenge. All of these demands require advances in materials engineering and metal processing technology, although there have been many breakthroughs in anodic oxidation technology, micro-arc oxidation technology, covalent grafting technology, and layer-by-layer deposition technology. However, exploration by metal engineering scholars is still needed. Each part of this review describes in detail the difficulties faced by each achievement, as well as the solutions of existing metal processing technologies, so we will not repeat them here. The comparison of various titanium implant processing techniques is also underway; for example, some studies have reported that the efficiency of surface modification of Ti-based implants is not as effective as surface roughening [121,122].

However, it is worth mentioning that, after implantation, the interaction between the titanium implant and the human body will produce various complex biological, physiological, and chemical reactions. In addition to the antibacterial, antitumor, and osseointegration capabilities of titanium implants—which are the most important in the field of orthopedics—the better biocompatibility of titanium implants is also an indispensable characteristic. RF magnetron sputtering, plasma spraying, and other technologies also play an important role in the preparation of titanium implants; in some reports, achievements have also been made in enhancing the biocompatibility of titanium implants [123,124,125,126]. In summary, in orthopedic clinics, in order to meet the needs of orthopedics, the processed titanium-based implants should have good physical properties, including elastic modulus, fatigue resistance, etc.; biological properties, including antibacterial properties, osseointegration capabilities, biocompatibility, etc.; and other necessary characteristics, including drug-carrying capacity and corrosion resistance. The processing technology of titanium implants should balance these characteristics of the implants, so that the necessary indicators can achieve satisfactory results; however, this is a source of difficulty in this field, and further research and exploration are still needed.

However, from another perspective, some innovative attempts are gradually emerging. Some researchers are dedicated to processing a variety of Ti-based alloys; they use the various techniques mentioned above to obtain titanium substrates loaded with different chemical compositions or organics, instead of simply physically modified titanium [127]. This seems to open up a whole new research direction. However, considering the clinical perspective of orthopedics, the focus should be on how to ensure the biological safety, versatility, and effectiveness of these chemical components and loaded drugs.

### 5.4. Clinical Application of Titanium Implants

In orthopedic clinics, it is also necessary to provide personalized titanium implants for patients with different needs in order to achieve the purpose of treatment. The patients who need implant surgery are usually patients who have undergone massive bone resection of bone tumors, patients with severe osteomyelitis, patients with bone defects caused by trauma, etc. As for patients with bone tumors, titanium implants carrying antitumor drugs (doxorubicin, cisplatin, curcumin, etc.) are usually used. Titanium implant scaffolds are also required for better osseointegration efficiency, because of their antitumor effects. Long-term effective local delivery of drugs and rapid osseointegration achieve better results. The combination of chemotherapy drugs and surgery has become one of the most important methods of bone tumor treatment. As for patients with severe osteomyelitis, because the tissue surrounding the titanium implant is already in a state of bacterial infection, delayed union or nonunion of the bone is very likely to occur. Orthopedic practitioners usually choose to load implants with multiple antibiotics (vancomycin, gentamicin, etc.). Coating-loaded titanium implants, combined with antibiotic bone cement and other materials for treatment, use the continuous and slow release of antibiotics around the titanium implants to keep the local antibiotic levels high, which is beneficial to the recovery of patients with osteomyelitis. As for patients with large bone defects caused by trauma, if the distance between the fracture ends exceeds 1–2 cm, they cannot heal by themselves. Orthopedic practitioners usually choose titanium implants loaded with bone anabolic drugs, anti-catabolic drugs, or growth factors (rhBMP2, zoledronate, simvastatin, calcitriol, etc.). At the same time, titanium scaffold implants are required to have suitable physical properties (greater porosity, higher roughness, etc.) in order to assist the slow local delivery of drugs to better promote bone growth and osseointegration.

It cannot be ignored that it is very meaningful to explore the effects of local loading of antitumor drugs combined with systemic chemotherapeutics on titanium implants at different tumor stages; however, there are very few relevant studies at present. We believe that the reason for this issue is that the number of patients is small (the number of patients with bone tumors is small), the severity of tumors is different, the sensitivity to individual drugs is different (whether in Homo sapiens or experimental animals), and there are constraints posed by relevant ethical laws. It is very difficult to organize a large sample for research, and it is necessary to overcome the various problems mentioned above. Therefore, the synergistic antitumor effect of systemic chemotherapy and local drug delivery systems has no accepted conclusion, and requires a large number of long-term studies.

## 6. Conclusions

With the development of pharmacology and pharmacokinetics, the drug industry is constantly innovating to adapt to various implant environments. At the same time, with the development of metal materials science, the processing technology of titanium implants is gradually improving, and various devices that conform to human biomechanics and are better able to store and slowly release drugs have been prepared, allowing drug delivery systems to achieve a longer action period and more stable drug delivery. In the future, more research should focus on the “holistic nature” of the synergy between titanium implants and their drug-loading systems, in order to improve the antibacterial properties of the implants, their ability to promote osseointegration, the balance of physical properties, and other personalized requirements, comprehensively solving the various required properties of implants. From the perspective of the field of orthopedics thus far, combined with the latest developments in pharmacology and metal materials science, we can synergistically solve more problems in the field of orthopedics.

## Figures and Tables

**Figure 1 nanomaterials-12-00047-f001:**
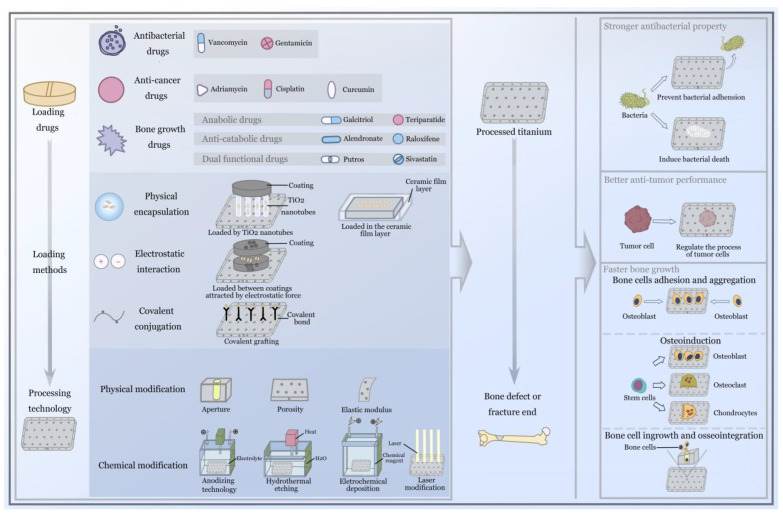
Schematical presentation of local drug delivery system on a titanium implant modified by metal processing technology, and the excellent advantages after implantation.

**Figure 3 nanomaterials-12-00047-f003:**
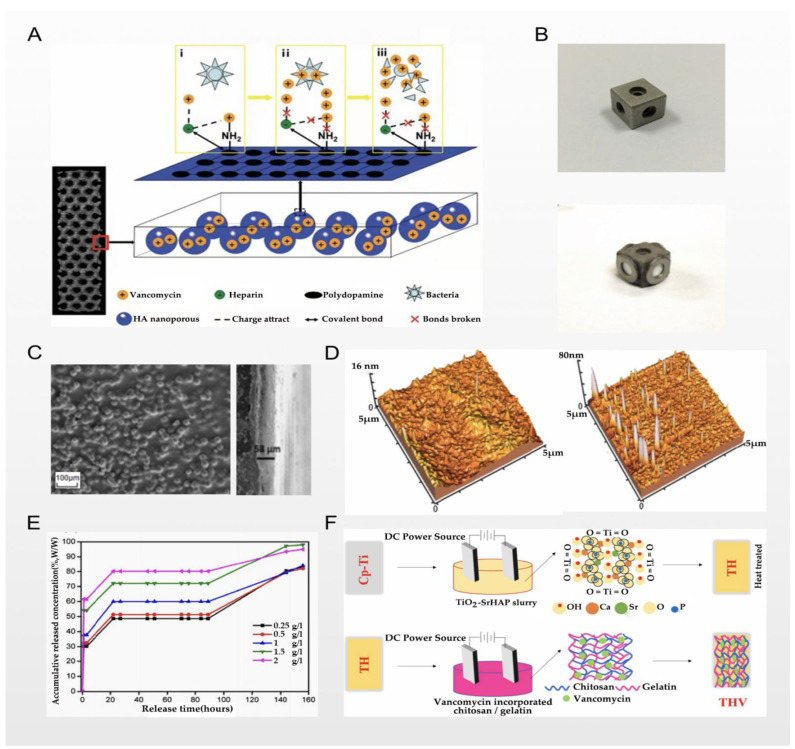
(**A**) Experimental scheme of Zhang et al.; reprinted with permission from [29]; copyright 2020 The Royal Society of Chemistry. (**B**) Intuitive structure of the designed material; reprinted with permission from [50]; copyright 2015 Martin B. Bezuidenhout et al. (**C**) Top-view and cross-sectional SEM images of the material; reprinted with permission from [30]; copyright 2014 Elsevier B.V. (**D**) Representative topographical AFM images of chitosan and a drug-eluting composite coating; reprinted with permission from [30]; copyright 2014 Elsevier B.V. (**E**) Drug release curve of experimental and control groups; reprinted with permission from [30]; copyright 2014 Elsevier B.V. (**F**) Experimental scheme of Nancy et al.; reprinted with permission from [58]; copyright 2018 Elsevier B.V.

**Figure 4 nanomaterials-12-00047-f004:**
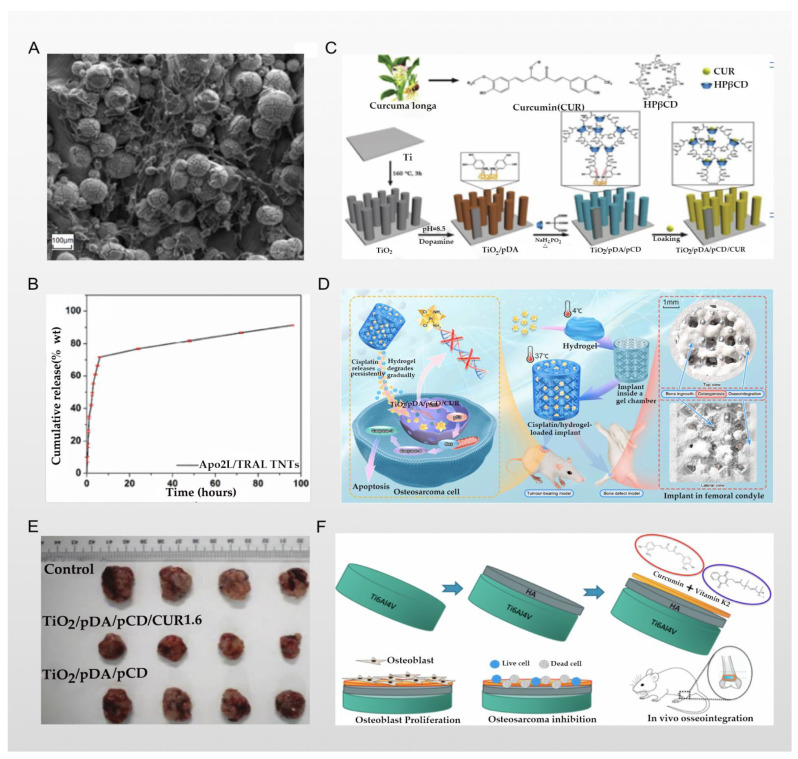
(**A**) Adhesion and aggregation of fibroblasts under an electron microscope; reprinted with permission from [31]; copyright 2017 American Chemical Society. (**B**) In vitro drug release of different drugs loaded onto TNT-3D-Ti implants (Apo2L/TRAIL); reprinted with permission from [31]; copyright 2017 American Chemical Society. (**C**) The experimental scheme of Zhang et al; reprinted with permission from [74]; copyright 2019 WILEY-VCH Verlag GmbH & Co. KGaA, Weinheim. (**D**) The experimental scheme of Jing et al; reprinted with permission from [76]; copyright 2021 Zehao Jing et al. (**E**) Images of tumors collected from tumor-bearing mice after various treatments; reprinted with permission from [74]; copyright 2019 WILEY-VCH Verlag GmbH & Co. KGaA, Weinheim. (**F**) The experimental scheme of Sarkar et al.; reprinted with permission from [77]; copyright 2020 American Chemical Society.

**Figure 5 nanomaterials-12-00047-f005:**
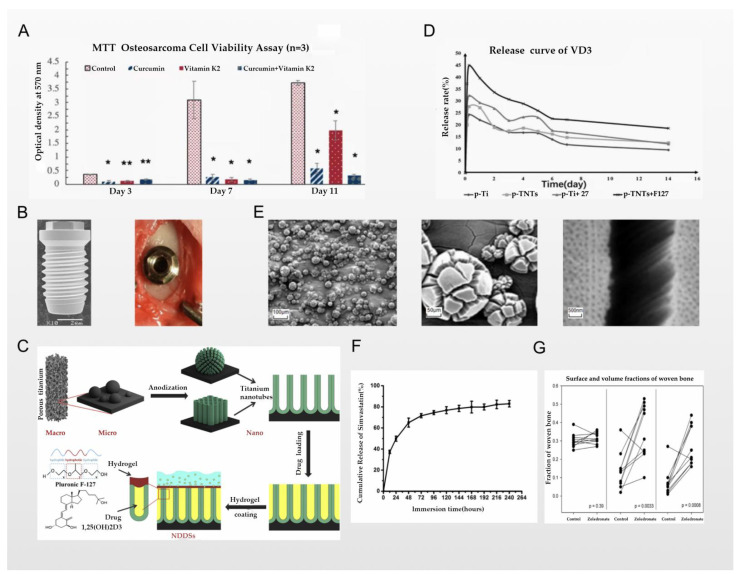
(**A**) MTT assay showing the effects of curcumin, vitamin K2, and curcumin + vitamin K2 on osteosarcoma cell viability (* denotes *p* ≤ 0.001, ** denotes *p* ≤ 0.05); reprinted with permission from [77]; copyright 2020 American Chemical Society. (**B**) Scanning electron microscopy image of the experimental implant at 10× magnification, and surgical placement of control; reprinted with permission from [81]; copyright 2017 John Wiley & Sons A/S. (**C**) The experimental scheme of He et al.; reprinted with permission from [82]; copyright 2019 Elsevier B.V. (**D**) The release curves of VD3 on samples; reprinted with permission from [82]; copyright 2019 Elsevier B.V. (**E**) Scanning electron micrograph (SEM) images of anodized 3D-printed Ti; reprinted with permission from [83]; copyright 2016 John Wiley & Sons, Ltd. (**F**) The cumulative release profile of SV from SV-LbL-coated Ti substrate in PBS; reprinted with permission from [84]; copyright 2018 Taylor & Francis. (**G**) Fractions of lamellar bone in contact with the implant surface and in a 0–1 mm zone around the implant; paired data are connected by a line; reprinted with permission from [85]; copyright 2017 Orthopaedic Research Society.

## Data Availability

Not applicable.

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
