# Peer review of "Titanium Implants and Local Drug Delivery Systems Become Mutual Promoters in Orthopedic Clinics"

_nanomaterials, 2021, doi:10.3390/nano12010047_

Round 1

Reviewer 1 Report

  1. Scheme 1 , the fonts are barely seen. Figure 1 and other figures as well. Please check that all important details are properly seen. In most of the figures it is not the case.
  2. It is known that implant surface modification is necessary to improve biocompatibility of implants. Please address some approaches briefly, if they are prospective ,e.g. plasma spraying. RF magnetron sputtering, org/10.1016/j.surfcoat.2021.127098, etc . In addition , some papers addressed that the efficiency of surface modification of Ti-based implants is not as effective as surface roughening.
  3. Titania nanotubes are deposited on a variety of Ti-based alloys and are prospective to be used for a variety of applications. Please address some important details reported elsewhere org/10.1016/j.msec.2018.12.045
  4. If the authors adopt figures they should provide details if permissions are obtained.
  5. Osseointegration of Ti-based implants is reported. Please address the issue of vascularization, which is also important for implant success.
  6. The trend is to use ceramics, which reveal better corrosion resistance compared with metals. Please provide why Ti-implants are still superior over another materials.

Author Response

Response to Reviewer 1:

Response: Thank you very much for your comments. We have revised the manuscript according to your suggestions. All the changes of the manuscript have been marked red and the response to the questions arisen from the review comments has been highlighted in Blue.

Comments and suggestions ①:

Scheme 1 , the fonts are barely seen. Figure 1 and other figures as well. Please check that all important details are properly seen. In most of the figures it is not the case.

Response: Thank you for your careful reminding. After re-uploading, we believe that the clarity of Figure and Scheme1 will be improved, and they meet the requirements of the publisher. If there is still a problem, I will actively cooperate with the editor to improve it.

Comments and suggestions ②:

It is known that implant surface modification is necessary to improve biocompatibility of implants. Please address some approaches briefly, if they are prospective ,e.g. plasma spraying. RF magnetron sputtering, org/10.1016/j.surfcoat.2021.127098, etc . In addition , some papers addressed that the efficiency of surface modification of Ti-based implants is not as effective as surface roughening.

Response: 

Thank you for your comment. Ensuring better biocompatibility of materials is an essential character for successful implantation. So your suggestions are very instructive and make our review more comprehensive. We have elaborated the article in your comment in our review On Page8, Line324-356, the added information is:

Furthermore, Chernozem et al. reported that while combining the above-mentioned anodization and electrochemical deposition techniques, they paid more attention to improving the biocompatibility of titanium implants, which was also beneficial to improving the antibacterial properties of titanium implants. In order to balance the various complex reactions after implantation. To improve the biocompatibility, Chernozem et al. prepared anodized TiO2 NTs surface, and then used electrochemical deposition technology to deposite synthesize Ag NPs and CaP NPs on the TiO2 NTs surface. Since the surface of TiO2 NTs is hydrophilic and the application of Ag NPs leads to a decrease of the water CA and an increase of the surface free energy due to increased contribution of the polar component, whereas the surface of biocom-posites with CaP NPs is superhydrophilic. The characteristics of the above titanium implants lead to its better biocompatibility and antibacterial properties, which have also been verified in subsequent cell experiments. At the same time, they demostrate that fabrication of Ag and CaP NPs which inhibit the growth of bacteria and can be used for functionalization of titania NTs【https://doi.org/10.1016/j.msec.2018.12.045】. Chernozem et al. also emphasized the importance of biocompatibility on the basis of enhancing the antibacterial properties of implants. Generally speaking, biocompatibility refers to the degree of mutual acceptance of materials, living tissues and body fluids, that is, degree of foreign body reaction. The current research on the biocompatibility of titanium implant materials mainly focuses on the following three aspects: ①The over-all physiological impact of titanium implants on tissues and organs. ②The metabolic process of the degradable part of titanium implants in the body. ③ The effect of titanium implants on information transmission and gene regulation among cells, tissues and organsDOI:10.22203/eCM.v039a16 DOI: 10.1016/j.msec.2015.09.059 DOI: 10.1016/j.jmbbm.2020.103671

】. The molecular composition and structure of the surface of the biometal material strongly affect the composition and structure of the surface of the biometal material strongly affect the composition and structure of the protein it adsorbs, so its subtle changes can significantly change the biological activity of the material. The material can be modified by the surface modification of titanium implants and other processing techniques. The surface is effectively controlled. However, the current research is mainly focusing on ① (as the report of Chernozem et al.), but there are few researches on ② or ③. The biocompatibility of titanium implants requires deeper exploration in the future【10.3389/fbioe.2020.576969】.

Not only that, but further discussion was conducted in the Processing Technology of Titanium Implants section in dis. At present, magnetron sputtering technology and plasma spraying technology also play a pivotal role in the preparation of titanium-based materials [DOI: 10.1016/j.msec.2019.01.099DOI: 10.1007/s10856-020-06477-4DOI: 10.1016/s0142-9612(03)00067-xDOI: 10.1021/acs.langmuir.1c00411DOI: 10.1016/j.actbio.2018.11.041]. which highlights the important position of titanium-based materials to ensure their biocompatibility during the preparation, and is also one of the key goals of metal implant design. In the field of orthopedics, it can cooperate with loaded drugs to achieve better therapeutic effects. As for the processing technology of titanium-based implants, some studies reported that the efficiency of surface modification of Ti-based implants is not as effective as surface roughening [DOI: 10.3390/jcm9082627DOI: 10.3390/ma13010089]. Indeed, the correct process choice is to give titanium implants their own characteristics. Basically, this content is also discussed in the Processing Technology of Titanium Implants section in Dis. On page21, line842-864, the added information is:

But it is worth mentioning that, after implantation, the interaction between the titanium implant and the human body will produce various complex biological, physiological and chemical reactions. In addition to the antibacterial, anti-tumor and osseointegration ability of titanium implants that are the most important in the field of orthopedics, the better biocompatibility of titanium implants is also an indispensa-ble character. RF magnetron sputtering technology, Plasma spraying technology and other technology also play an important role in the preparation of titanium implants. In some studies, achievements have also been made to enhance the biocompatibility of titanium implants [DOI: 10.1016/j.msec.2019.01.099DOI: 10.1007/s10856-020-06477-4DOI: 10.1016/s0142-9612(03)00067-xDOI: 10.1021/acs.langmuir.1c00411DOI: 10.1016/j.actbio.2018.11.041]. In summary, in orthopedics clinics, in order to meet the needs of orthopedics, the processed titanium-based implants should have physical properties including elastic modulus, fatigue resistance, etc.; biological properties including antibacterial properties, osseointegration capabilities, and compatibility, etc.; other necessary characteristics including drug carrying capacity and corrosion resistance. The processing technology of titanium implants should balance these characteristics of the implants, so that the necessary indicators can achieve satisfactory results. However, this is also the focus and difficulty in this field, and further research and exploration are still needed.

Comments and suggestions ③:

Titania nanotubes are deposited on a variety of Ti-based alloys and are prospective to be used for a variety of applications. Please address some important details reported elsewhere org/10.1016/j.msec.2018.12.045.

Response:  Thank you for your comment, the variety of Ti-based alloys should indeed be considered in our review, which will make the review more comprehensive regarding the processing technology of titanium implants. However, it is more meaningful to explore the relationship between loaded drugs and pure titanium implants(Usually physical modification) from the perspective of extensive clinical applications based on orthopedics: We believe that when commercially available drugs and simply processed pure titanium substrates work in synergy, this will be more biosafe, universal, and effective. At the same time, we also searched related articles, and it can be concluded that most researchers will load the drug on the surface of the titanium implant with simple modification. However, with different chemical processing titanium (such as CaP, graphene, etc.), these chemical components may affect the pharmacokinetic properties and the drug effect. So we chose the former for extensive discussion. But after your valuable suggestions, we summarized the article you mentioned in the Processing Technology of Titanium Implants section in Dis, and made a detailed explanation. the added information is:On page21, line858-864

From another perspective, some innovative attempts are gradually emerging. Some researchers are dedicated to processing variety of Ti-based alloys. They use various techniques mentioned above to obtain titanium substrates loaded with different chemical compositions or organics instead of simply physically modified titanium [//doi.org/10.1016/j.surfcoat.2021.127098 ] , This seems to open up a whole new research direction. However, considering the clinical perspective of orthopedics, how to ensure the biological safety, versatility and effectiveness of these chemical components and loaded drugs may be the focus.

Comments and suggestions ④:

If the authors adopt figures they should provide details if permissions are obtained.

Response: Thank you for your kind reminder. I have sorted out the details of each quoting figure below and their references and I have submitted the permission file of the Figures.

Comments and suggestions ⑤:

Osseointegration of Ti-based implants is reported. Please address the issue of vascularization, which is also important for implant success.

Response: Thank you for your introductory comments. In our article, we discussed more about cytological morphology issues, while ignoring issues at the tissue level (the reason for this problem may be that there are fewer articles on vascularization in this field) , Such as the vascularization you mentioned. Therefore, in the summary part of osseointegration (On page 15 Line561-574), we summarized the related issues of the mechanism and role of vascularization

In addition to the cytological mechanism (osteoblasts, osteoclasts, fibroblasts, macro-phages, neutrophils, etc.), the histological mechanism cannot be ignored. Accompanied by the regeneration of bone cells, the endothelial cells around the implant are regulated by angiogenesis activators or pro-angiogenic factors (eg. aFGF, bFGF, VEGF, etc.), and the vascular endothelial cells arranged along the blood vessel accelerate the prolifera-tion, The formation of new blood vessels. 【Biomed Mater Res A 2006;79:882–894. Biomaterials 2007;28:3679–3686.】Vascular invasion pro-vides transport of nutrients, wastes and precursor cells for growing/regenerating bone tissues. It also supports cross-talk between blood vessel endothelial cells and precursor cells to promote osteoblastic differentiation. Vascularization of peri-implant tissue is also very important to the remodeling and preservation of bone around an implant af-ter placement.【Endocrinology 2000;141:1667–1674. J Bone Miner Res 2005;20:2028–2035.】Therefore, the regeneration of bone cells at the cy-tological level and the formation of blood vessels at the histological level are used to construct nutrient transport channels, which will lead to better osseointegration effects.  On the other hand, the use of drugs to promote osseointegration has gradually been paid more attention by more scholars. Compared with the systemic administration of drugs to promote osseointegration before and after implantation, a local drug delivery system that carries osseointegration drugs (Promoting bone growth or angiogenesis drugs) on titanium implants has a better effect. This part will focus on the local drug delivery of titanium implants in collaboration with the commonly used clinical calcit-riol, indomethacin, simvastatin ,and bisphosphonates and VEGF to synergistically accelerate the progress of osseointegration. And the drug loading and release technology of the above-mentioned drugs.

At the same time, in the osseointegration-chemical coating section (on page21 Line706-733), we have added several references to the literature and discussions on related issues, the added information is: Leedy et al. reported that they designed a VEGF-loaded chitosan coating on the ti-tanium implant, and the chitosan was chemically bonded to the surface of the titanium implant through silane-glutaraldehyde. Vascular endothelial growth factor (VEGF), a heparin-binding growth factor specific for vascular endothelial cells, has a powerful growth factor that promotes angiogenesis and promotes osteoblast differentiation and bone regeneration. 【DOI: 10.1002/jbm.a.36559】 The release of VEGF is controlled by chitosan, and the release duration is about 14 days, which is divided into a burst release time of 12 hours, a plateau release of 3 days and a slow release of 11 days. The advantages of os-seointegration and vascularization have been confirmed in cell experiments. 【DOI: 10.1002/jbm.a.34745】 There is also a similar design in the report of Mullin et al. 【DOI: 10.1002/adhm.201700033】However, as a whole, there is less exploration of vascularization in this field, and this direction is worthy of follow-up research and exploration

Comments and suggestions ⑥:

The trend is to use ceramics, which reveal better corrosion resistance compared with metals. Please provide why Ti-implants are still superior over another materials.

Response: Thank you for your comment. Now ceramics is indeed the trend of choice of implant materials, because it has the characteristics of hydrophilicity and good affinity with biological tissues such as cells. We have listed other materials commonly used in orthopedics except titanium implants in int (On Page2 Line43-52), making the review more comprehensive. the added information is:

In these surgical operations, the use of implants has played an important role, mainly including metal implants (such as titanium alloys, stainless steel, chromium, nickel, tantalum, etc.), ceramics, and polymer materials (such as PEEK). They play a pivotal role in the field of orthopedics【DOI: 10.1002/jbm.a.36931】 In these surgical procedures, the use of implants plays a  pivotal role in the field of orthopedics[4]. Although each implant ma-terial has its own unique advantages, titanium alloy has become the most common metal implant due to its excellent biocompatibility, low elasticity and corrosion re-sistance. At the same time, as for the titanium implants loaded with drugs,there are more studies than other types of implants. Titanium alloys have become the most common mental implants due to its excellent biocompatibility, low elasticity and corrosion resistance.

We believe that none of these materials has an absolute advantage in the selection of orthopedic clinics. In this research, we focus on the synergistic relationship between implants and drugs, which has a larger number of articles, so we focus more on titanium-based materials. But as your reminder, the synergy of other materials and drugs should be the direction of future research. This is already in the added information.Thank you for your all comments again.

Reviewer 2 Report

The review presented in the paper is very useful and interesting.

Two remarks.

1. The short description of the first figure presented and named as "Scheme 1. A general overview of the content of the review." should be added in the beginning of Section unnumbered and named as "Local Drug Delivery System And Titanium Implant"

2. The Section "Conclusions" should be added.

Author Response

Response to Reviewer 2:

Comments: The review presented in the paper is very useful and interesting. 

Response: Thank you very much for your comments. We have revised the manuscript according to your suggestions. All the changes of the manuscript have been marked red and the response to the questions arisen from the review comments has been highlighted in Blue.

Comments and suggestions ①:

The short description of the first figure presented and named as "Scheme 1. A general overview of the content of the review." should be added in the beginning of Section unnumbered and named as "Local Drug Delivery System And Titanium Implant"

Response: Thank you for your careful reminder, we have made changes based on your reminder.

Comments and suggestions ②:

The Section "Conclusions" should be added.

Response: Thank you very much for your comment. We have added the content of the Dis section to make the discussion more comprehensive. The added content includes: more comprehensive processing technology of titanium implants (magnetron sputtering technology, plasma spraying technology, etc.); more comprehensive clinical problems; part of the molecular mechanism of osseointegration (histological level, Vascularization related issues) and many other issues.

the added information is:

Disccusion

From the clinical perspective of orthopedics, the development of local drug delivery systems on titanium implants has given orthopedic treatment a new, feasible and effective direction. This inevitably needs to balance the following four challenges: the pharmacological properties of loaded drugs, the molecular biological mechanism of bone cells and surrounding tissues, the processing technology of titanium implants and the correct choice of clinical application.

Loaded Drugs and Their Pharmacological Properties

In the clinical experience of orthopedics, the drugs frequently used with titanium implant surgery are mainly divided into several categories, analgesic and anti-inflammatory drugs (especially NSAIDs, acetaminophen and opioids are the most common), and some anti-bone tumors Drugs (doxorubicin, cisplatin and curcumin), antibiotics (vancomycin and clindamycin), anabolic drugs (calcitriol, teriparatide and fluvastatin), anti-catabolic drugs (alendronate sodium) , Zoledronate and raloxifene), dual anabolic and anti-catabolic drugs (simvastatin and strontium ranelate). Due to the pharmacological properties of these drugs, they are usually selected as systemic drugs or drugs loaded with titanium implants to achieve the purpose of personalized treatment. However, many reports have confirmed that the properties of these drugs sometimes affect titanium. The osseointegration effect of the implant interface, which deserves attention. NASAIDs control inflammation and pain by regulating the COX-2 pathway in arachidonic acid, thereby reducing the synthesis of prostaglandins, but Chikazu et al. and Zhao et al. reported that the lack of COX-2 can cause small adhesion and aggregation of osteoblasts around the mouse titanium implant is reduced, and the effect of osseointegration is weakened 112,113, but in the report of Yang et al., it is further believed that the controlled dose of NSAIDs has an effect. The key to the osseointegration effects 114. In response to the above problems, the advantages of acetaminophen are shown. It has a low effect on the COX-1 and COX-2 pathways. Although it is weak in inhibiting the synthesis of prostaglandins, it still exists. It has a certain anti-inflammatory and analgesic effect. It may be a correct choice for patients with non-severe pain and need to enhance the rate of osseointegration, which may be a hint for the subsequent choice of anti-inflammatory drugs for titanium implants effect. Teriparatide is a kind of anabolic drugs, which acts on the PTH receptor of osteoblasts, thereby affecting the level of parathyroid hormone to regulate bone production. However, when teriparatide is applied in large doses, it can cause osteoclasts. Extensive activation, and will cause the expression levels of specific transcription factors, osteocalcin, bone salivary protein and type I collagen in osteoblasts to be down-regulated to varying degrees, tending to bone catabolism. These conclusions are corroborated in the reports of Almagro et al. and Tao et al 115,116. Therefore, after the titanium implant is loaded with teriparatide, it is necessary to strictly control the local concentration of the drug around the implant. Zoledronate is an excellent anti-catabolic drug, which has a high affinity for mineralized bone. It targets farnesyl pyrophosphate synthase in osteoclasts, and then inhibits the activity of osteoclasts or directly Induces the apoptosis of osteoclasts to inhibit the bone resorption of osteoclasts 117, so scholars usually choose them as loading drugs for titanium implants. However, according to reports by Basudan et al. and Cardemil et al., zoledronate has different osseointegration effects on bone tissues in different parts of the human body. For example, the mandible and tibia and fibula have negative effects. Therefore, the application of zoledronic acid in different implant sites needs to be individually adjusted. The pharmacological properties of the drug itself and titanium implants may interfere with each other. Nowadays, there is relatively little exploration in this field. For example, whether anti-tumor drugs, opioids, etc. will affect the properties of titanium implants, and then Affect the osseointegration effect? It is necessary to be more cautious in the choice of implant loading drugs, and more future experiments are needed. In addition, scholars have paid more and more attention to melatonin in the treatment of bone tumors in recent years, but the use of melatonin in the local drug delivery of titanium implants is rare 118.

Molecular Biological Mechanism of Bone Cells and surrounding tissues

From the perspective of molecular biology of osteoblasts, osteoclasts, fibroblasts and mesenchymal stem cells, the mechanisms of bone aggregation, osteoinduction, and osseointegration on titanium implants have been deeply studied119-122. However, in recent years, there have been more and more explorations on the role of integrins on the surface of bone cells. Olivares-Navarrete et al. reported that they prepared titanium implants attached with a graphite carbon coating and studied the important role of integrins in the process of bone cells maturation and osseointegration on the surface. Integrin is a transmembrane heterodimer, and each subunit has its unique role. It is because of its existence that osteoblasts and mesenchymal stem cells are connected with extracellular proteins, including actin, Laminin, etc., so it regulates the subsequent important processes of bone cell attachment, aggregation, and migration. The experimental results of this study also indicate that the integrin α subunit plays a major role in surface chemical recognition. And within a certain range, the roughness of the titanium implants can enhance the effect of bone differentiation and osseointegration123. The exploration from the perspective of molecular biology also provides instructions for the selection and application of implants. For example, due to the surface chemical recognition of integrin α subunits, it is possible to choose to loading drugs or other agents to target the integrin α subunit so that affect the maturation of osteoblasts. It is also possible to reasonably increase the surface roughness of the titanium implant to enhance the osseointegration effect. In order to play a prompting role, the molecular biological mechanisms of bone growth, osteoinduction, and osseointegration on the surface of titanium implants still need to be further explored. In addition, as for the related issues of vascularization at the histological level (relevant mechanisms have been elaborated in 4.1), there are few researches and few related commercial drugs, which is expected to become a key research direction in the future.

Processing Technology of Titanium Implants

The physical modification of titanium implants, surface modification and other processing technologies play an important role in its own antibacterial properties and the ability to promote osseointegration. It has summarized that the predecessors prepared the TiO2 nanotube structure to carry drugs through different metal processing techniques such as the micro/nano-level modification technology of the titanium surface, and adjusted the diameter, vertical depth and tube wall thickness of the titanium nanotubes to control the drug Load or its release, preparation of ceramic thin film layers by electrochemical deposition to store drugs And by preparing a coating structure on the surface of the implant, using the ability of the coating structure to load the drug to load and control the release of the drug. Generally speaking, the common challenges focus on how to solve the stability of drug loading and long-term slow drug release, and whether the physical properties of titanium implants, such as elastic modulus, corrosion resistance, and fatigue resistance, can meet the needs of patients. At the same time, under different demand conditions, the requirements for the pore size, porosity and internal spatial structure of titanium implants (titanium scaffolds) are also different. Under this complex structure, the control of drug loading and release has become new challenges. All these require advances in materials engineering and metal processing technology, although there have been many breakthroughs in anodic oxidation technology, micro-arc oxidation technology, covalent grafting technology and Layer-by-Layer technology. However, the exploration of metal engineering scholars is still needed. Each part of the review describes in detail the difficulties faced by each achievement and the solutions of existing metal processing technologies, so it won't repeat them here. The comparison of various titanium implant processing techniques is also underway, for example, some studies reported that the efficiency of surface modification of Ti-based implants is not as effective as surface roughening124,125.

But it is worth mentioning that, after implantation, the interaction between the titanium implant and the human body will produce various complex biological, physiological and chemical reactions. In addition to the antibacterial, anti-tumor and osseointegration capabilities of titanium implants that are the most important in the field of orthopedics, the better biocompatibility of titanium implants is also an indispensable character. RF magnetron sputtering technology, Plasma spraying technology and other technology also play an important role in the preparation of titanium implants,in some reports, achievements have also been made to enhance the biocompatibility of titanium implants 126-129. In summary, in orthopedics clinics, in order to meet the needs of orthopedics, the processed titanium-based implants should have physical properties including elastic modulus, fatigue resistance, etc.; biological properties including antibacterial properties, osseointegration capabilities, and compatibility, etc.; other necessary characteristics including drug carrying capacity and corrosion resistance. The processing technology of titanium implants should balance these characteristics of the implants, so that the necessary indicators can achieve satisfactory results. However, this is also the focus and difficulty in this field, and further research and exploration are still needed.

However, from another perspective, some innovative attempts are gradually emerging. Some researchers are dedicated to processing variety of Ti-based alloys. They use various techniques mentioned above to obtain titanium substrates loaded with different chemical compositions or organics instead of simply physically modified titanium 130, This seems to open up a whole new research direction. However, considering the clinical perspective of orthopedics, how to ensure the biological safety, versatility and effectiveness of these chemical components and loaded drugs may be the focus.

Clinical Application of Titanium Implants

In orthopedic clinics, it is also an art to provide personalized titanium implants for patients with different needs to achieve the purpose of treatment. The patients who need implant surgery are usually patients who have undergone massive bone resection of bone tumors, patients with severe osteomyelitis,  patients with bone defects caused by trauma and so on. As for patients with bone tumors, titanium implants carrying anti-tumor drugs (doxorubicin, cisplatin, curcumin, etc.) are usually used. Titanium implant scaffolds are also required for better osseointegration efficiency because of their anti-tumor effects. Long-term effective local delivery of drugs and rapid osseointegration achieve better results. Combination of chemotherapy drugs and surgery has become one of the important methods of bone tumor treatment. As for patients with severe osteomyelitis, because the tissue surrounding the titanium implant is already in a state of bacterial infection, delayed or nonunion of the bone is very prone to occur. Orthopedics usually choose to carry multiple antibiotics (vancomycin, gentamicin, etc.) Coating-loaded titanium implants, combined with antibiotic bone cement and other materials for treatment, use the continuous and slow release of antibiotics around the titanium implants to keep the local antibiotic level at a high level, which is beneficial to patients with osteomyelitis for recover. As for patients with large bone defects caused by trauma, if the distance between the fracture ends exceeds 1-2cm, they cannot heal by themselves. Orthopedics usually choose titanium implants loaded with  bone anabolic drugs, anti-catabolic drugs or growth factors(rhBMP2, zoledronate, simvastatin, calcitriol, etc.). At the same time, titanium implant scaffolds are required to have suitable physical properties (larger porosity, higher roughness, etc.) to assist the slow local delivery of bone growth drugs to better promote bone growth and osseointegration.

It cannot be ignored that it is very meaningful to explore the effect of local loading of anti-tumor drugs combined with systemic chemotherapeutics on titanium implants at different stages of the tumor. However, there are very few relevant studies at present. We believe that the reason for this issue is that the number of patients is small (the number of patients with bone tumors is small), the severity of tumors is different, the sensitivity of individual drugs is different (whether Homo sapiens or experimental animals), and the constraints of relevant ethical laws, etc. It is very difficult to organize a large sample of research and it is necessary to overcome the various problems mentioned above. Therefore, the anti-tumor synergistic effect of systemic chemotherapy and local drug delivery system has no accepted conclusion, which requires a large number of long-term studies.

With the development of pharmacology and pharmacokinetics, the drug itself is constantly innovating to adapt to various implant environments. At the same time, with the development of metal materials science, the processing technology of titanium implants is gradually innovating, and various devices that conform to human biomechanics and are more ingenious to store and slow-release drugs have been prepared. This allows the drug delivery system to have a longer action time and a more stable drug output. In the future, more research should focus on the "holistic nature" of the synergy between titanium implants and their drug-loading systems, to achieve the antibacterial properties of the implants, the ability to promote osseointegration, the balance of physical properties and other personalized requirements, and comprehensively Solve the various required properties of implants. On the basis of today, combined with the latest developments in pharmacology and metal materials science, we can synergistically solve more problems in the field of orthopedics.

Reviewer 3 Report

The present review seems to be interesting due to the clinical relevance of the topic in orthopedics. Titanium implants are commonly used as the treatment of choice, where a local release of drugs provides many advantages over the use of systemic drugs. In this sense, this work aims to analyze the current state of local drug delivery systems that cooperate with titanium implants to enhance antibacterial, anti-tumor and osseointegration effects.

  1. Introduction

The introduction highlights the importance of the use of titanium implants in situations of diseases that seriously affect bone structure and function. The biocompatibility, low elasticity and resistance to corrosion of this material allows osseointegration, key to the success of implants in the treatment of orthopedic diseases. The administration of local versus systemic drugs is a great advantage by improving the drug concentration in the peri-implant tissues and improving the rate of biological utilization. However, the authors do not justify why they only focus on analyzing antibacterial, antitumor and bone integration improvement treatments.

  1. Antibacterial

The authors show the mechanisms by which bacteria lead to osseointegration problems. However, they should explain the mechanisms by which bacterial contamination occurs, paying special attention to the influence of the main route of contamination (through the surgical wound) with respect to the local administration of antibiotics.

Only the use of vancomycin is discussed. This fact must be justified.

Is the use of vancomycin useful for Pseudomonas aeruginosa, common pathogens in titanium implant infections?

  1. Anti-tumor

Patients with resections of bone tumors are usually treated with titanium implants. The implantation of titanium implants with anti-bone tumor drugs can be a good option to achieve an effective concentration around bone tumors, avoiding the toxicity of systemic chemotherapy drugs. Due to the severity of the pathology, the authors must address the efficacy of this unique practice or its need for combination with systemic therapy.

  1. Osseointegration Effect

Osseointegration is the biological procedure by which bone adhesion occurs in close contact with the implant surface. The authors present different substances and surface procedures that appear to improve osseointegration. However, the time of this osseointegration process and its relationship with the local release of substances must be taken into account in a time-limited manner.

  1. Discussion

Although the authors make an adequate analysis of what is addressed in the review, this resembles more the comments of the authors that a discussion supported or contrasted with the existing bibliography.

In general, in addition to an excessively broad review of the local drug delivery systems literature on titanium implants, it does not provide news, also using 50% of bibliographic references older than 5 years.

Author Response

Response to Reviewer 3:

The present review seems to be interesting due to the clinical relevance of the topic in orthopedics. Titanium implants are commonly used as the treatment of choice, where a local release of drugs provides many advantages over the use of systemic drugs. In this sense, this work aims to analyze the current state of local drug delivery systems that cooperate with titanium implants to enhance antibacterial, anti-tumor and osseointegration effects.

Response: Thank you very much for your comments. We have revised the manuscript according to your suggestions. All the changes of the manuscript have been marked red and the response to the questions arisen from the review comments has been highlighted in Blue.

Comments and suggestions ①:

Introduction

The introduction highlights the importance of the use of titanium implants in situations of diseases that seriously affect bone structure and function. The biocompatibility, low elasticity and resistance to corrosion of this material allows osseointegration, key to the success of implants in the treatment of orthopedic diseases. The administration of local versus systemic drugs is a great advantage by improving the drug concentration in the peri-implant tissues and improving the rate of biological utilization. However, the authors do not justify why they only focus on analyzing antibacterial, antitumor and bone integration improvement treatments.

Response: Thank you for your comment. Indeed, explaining why only focusing on these three aspects will help readers to grasp the overall content of the review more clearly. From the perspective of orthopedic surgeons, we choose the abilities of titanium implants according to clinical needs(and other articles). Reviewing the entire treatment process, 1. In order to ensure the success of implantation, the most fundamental thing is to avoid bacterial infection of the implant and reduce foreign body reaction at the same time(antibacterial and biocompatibility). 2. After successful implantation, the problem of better osseointegration efficiency in patients with fractures or bone defects need to be solved. (Osseointegration effect) 3. After achieving successful bone ingrowth, solve the primary diseases that cause fractures or bone defects such as bone tumors and osteomyelitis(anti-tumor or antibacterial). Therefore, the biocompatibility, antibacterial, anti-tumor and osseointegration capabilities of titanium-based implants are the most required properties of titanium implants. Although the required properties include corrosion resistance, appropriate elastic modulus, fatigue life, etc., although implants are indispensable to ensure these excellent properties, clinically, more attention is paid to the above-mentioned properties. At the same time, the number of relative articles is greater. However, there are currently few studies exploring the improvement of the biocompatibility of drugs carried by titanium implants, and there is a lack of commercialized drugs to improve biocompatibility, so we have integrated this part into the antibacterial section.The added information is(On Page19 Line148-160):

As for a titanium-based implant in orthopedic applications, in clinical practice, orthopedic surgeons usually give more consideration to these aspects empirically. 1. In order to ensure the success of implantation, the most fundamental thing is to avoid bacterial infection of the implant and reduce foreign body reaction. 2. After successful implantation, the problem of better osseointegration efficiency in patients with fractures or bone defects need to be solved. 3. After achieving successful bone ingrowth, solve the primary diseases that cause fractures or bone defects such as bone tumors and osteomyelitis. So in the whole process of this treatment, the biocompatibility, antibacterial, anti-tumor and osseointegration capabilities of titanium-based implants are the most required properties. This review will focus on the synergy of local drug delivery systems and titanium implants to solve key clinical issues such as antibacterial (The issues related to biocompatibility will be explained in the section on antibacterial properties), anti-tumor and osseointegration effects in recent years.

Comments and suggestions ②:

Antibacterial

  • The authors show the mechanisms by which bacteria lead to osseointegration problems. However, they should explain the mechanisms by which bacterial contamination occurs, paying special attention to the influence of the main route of contamination (through the surgical wound) with respect to the local administration of antibiotics.

2,Only the use of vancomycin is discussed. This fact must be justified.

3,Is the use of vancomycin useful for Pseudomonas aeruginosa, common pathogens in titanium implant infections?

Response1: Thank you for your comment. We have summarized several important ways of bacterial invasion in orthopedics clinics, the added information is: On page 4 Line162-175

  • Although orthopedic surgeons pay great attention to aseptic surgical operations, there is still the possibility of bacterial invasion. In orthopedic clinics, the bacterial invasion of titanium implants is usually due to these aspects, 1. For example, bacterial invasion caused by open trauma, the bacteria remain in the epidermis, subcutaneous tissue or deep tissue, and it is usually difficult to completely eliminate all bacteria; 2. The patient’s own diseases such as bacteremia, bacteria adhere to the surface of the titanium implant through the blood circulation; 3, the operation that does not meet the requirements during the operation leads to the invasion of bacteria, or the infection of the incision after the operation causes the invasion of bacteria; 4, bacterial invasion has already occurred during the preparation or transportation of the titanium implant. In short, once a bacterial invasion occurs, it will occur in the deep tissues around the titanium implant, so that failed implantation will be inevitable, and the effect of disinfection and systemic antibiotics will be minimal【DOI: 1016/j.addr.2012.03.015DOI: 10.1097/BRS.0000000000003218DOI: 10.1126/sciadv.abb0025】.

The antibacterial property discussed in our review is a property of titanium-based implants. Through the implants carrying some antibiotics and other antibacterial agents, the antibacterial and sterilization environment around the implants is ensured, and bacteria colonization and formation are difficult to remove. The biofilm is protected from bacterial infection and implantation failure. However, bacterial attachment and infection of the epidermis and deep subcutaneous tissues at the surgical incision requires intraoperative aseptic operation and antibiotic treatment before and after surgery. These are two different research directions. We have explained these issues clearly because of your careful reminder.

Response2:Thank you very much for your comment. Due to the wide variety of antibiotics, although they have different pharmacological properties, the relationship between multiple antibiotics and the types of bacterial invasion of titanium-based implants is bound to be an important issue, and your suggestions are very important. We have elaborated on related issues in the antibacterial section, the added information is: (On Page 5 Line207-216)

As for the content we reviewed, it is precisely for the antibacterial properties of titanium-based implants that its drug delivery system is loaded with these antibiotics. Obviously, the load of these antibiotics was completed before surgery. Due to the broad-spectrum antibacterial properties of vancomycin, the type of bacterial invasion after implantation is unknown. When a choice must be made, vancomycin becomes the first choice [DOI: 10.5435/JAAOS-D-16-00033]. There is also an interesting fact: in the articles in this field from 2019 to 2021, most authors choose vancomycin for follow-up experiments.Therefore, we chose vancomycin as a representative to conduct a review in this field, but gentamicin, first-generation cephalosporin, etc., have also been reported by scientific researchers【41-43】.

Response3:Thanks for your comment, Vancomycin is still sensitive to Pseudomonas aeruginosa【DOI: 10.1186/s13756-018-0370-9】. The load of vancomycin is usually before implantation, so  broad-spectrum vancomycin was chosen to solve the entire perioperative infection and stabilize the implantation after the operation issues. This is just like in orthopedic clinics, when we have not found a clear pathogenic bacteria, based on experience, more vancomycin is used for antibacterial. Therefore, although it is not the best choice to choose vancomycin for the treatment of Pseudomonas aeruginosa during systemic antibacterial treatment. It is the best choice for titanium implants loaded with drugs. Thank you again for your comment again, we have elaborated relative issues in review. The added information is (Page 5 Line191-204):

The load of vancomycin is usually before implantation, so (broad-spectrum) vancomycin was chosen to solve the entire perioperative infection and stabilize the implantation after the operation issues. Although it is not the best choice to choose vancomycin for the treatment of Pseudomonas aeruginosa during systemic antibacterial treatment. It is the best choice for titanium implants loaded with drugs.

Comments and suggestions ③:

Anti-tumor

Patients with resections of bone tumors are usually treated with titanium implants. The implantation of titanium implants with anti-bone tumor drugs can be a good option to achieve an effective concentration around bone tumors, avoiding the toxicity of systemic chemotherapy drugs. Due to the severity of the pathology, the authors must address the efficacy of this unique practice or its need for combination with systemic therapy.

Response:Thank you for your instructive comments. It is very meaningful to explore the effects of local loading of anti-tumor drugs combined with systemic chemotherapeutics on titanium implants at different stages of the tumor. But after a long investigation and retrieval of relevant literature, unfortunately, such research is very rare. This is also an issue that we are particularly troubled. We believe that the reason for this issue is that the number of patients is small (the number of patients with bone tumors is small), the severity of tumors is different, the sensitivity of individual drugs is different (whether Homo sapiens or experimental animals), and the constraints of relevant ethical laws, etc. And it is very difficult to organize a large sample of research and it is necessary to overcome the various problems mentioned above. As for animal and cell experiments, only a few related studies have been described in the anti-tumor section. Therefore, the anti-tumor synergistic effect of systemic chemotherapy and local drug delivery system has no recognized conclusion, which requires a long-term study. However, it is certain that titanium implants loaded with anti-tumor drugs still have advantages in avoiding serious side effects caused by systemic chemotherapy. For example, after massive bone resection of bone tumors, implantation of titanium loaded with anti-tumor drugs can treat bone defects while inhibiting the progress of tumor cells locally to achieve better therapeutic effects. In summary, in the Clinical Application of Titanium Implants section of Discussion, we briefly describe the issues discussed above. the added information is(On Page22 Line891-901 ):

It cannot be ignored that it is very meaningful to explore the effect of local loading of anti-tumor drugs combined with systemic chemotherapeutics on titanium implants at different stages of the tumor. However, there are very few relevant studies at present. We believe that the reason for this issue is that the number of patients is small (the number of patients with bone tumors is small), the severity of tumors is different, the sensitivity of individual drugs is different (whether Homo sapiens or experimental animals), and the constraints of relevant ethical laws, etc. It is very difficult to organize a large sample of research and it is necessary to overcome the various problems mentioned above. Therefore, the anti-tumor synergistic effect of systemic chemotherapy and local drug delivery system has no accepted conclusion, which requires a large number of long-term studies.

Comments and suggestions ④:

Osseointegration Effect

Osseointegration is the biological procedure by which bone adhesion occurs in close contact with the implant surface. The authors present different substances and surface procedures that appear to improve osseointegration. However, the time of this osseointegration process and its relationship with the local release of substances must be taken into account in a time-limited manner.

Response: Thank you for your instructive suggestions, which is the issuses our research group is addressing. Indeed, after implantation, the release of the drugs under the control of the implant preparation technology and the osseointegration process of the surface bone aggregation will begin at the same time. It is bound to influence each other, and the release of the drugs for promoting bone regeneration and osseointegration will make the relative cells aggregation, adhesion, regeneration, and integration process on the surface of the titanium implant and enhanced by the action of the drug; At the same time, due to the gradual formation of bone trabecula, tissue vascularization, an increase in the number of bone cells that grow into the titanium alloy pores, and bone mineralization, it is bound to cause significant bone microenvironment around the drug-loaded titanium implant changes, which will have a certain impact on the process of drug release, diffusion, and absorption into the blood. Conversely, the above process causes the concentration change around the titanium implant to also affect the healing effect of the bone. In summary, in 4.1-4.2, we summarized in detail the relationship between the controlled release time of the drugs and the time of initial fracture healing in these articles (if relevant results exist in the experimental results) and the time of initial fracture healing (X-ray shows continuous callus at the fracture, and the fracture line is blurred). However, it is still facing difficulties. The relationship between the release of drugs and the process of osseointegration is more complicated. This is related to individual factors (Homo sapiens or laboratory animals), the type of drugs, the modification process of titanium implants, etc., which require an index to judge. The purpose is to balance the drug release and osseointegration effect of the titanium implant. It is still a problem to be solved in the future. The above issues have been discussed in detail in 4.1(On Page18 Line717-733 ) the added information is:

  • In summary, there is an issue that has to be taken into account, after implantation, the release of drugs under the control of implant preparation technology and the pro-cess of bone aggregation and osseointegration on the surface will begin at the same time, and they are bound to affect each other. The release of drugs that promote bone regeneration and promote osseointegration will make titanium implants the bone on the surface of the implant aggregates, adheres, and regenerates. The process of osseoin-tegration is enhanced by the action of drugs【DOI: 1111/clr.13602】; At the same time, with the gradual formation of bone trabeculae, tissue vascularization, an increase in the number of bone cells that grow into the titanium alloy pores, and bone mineraliza-tion, it is bound to lead to the development of the bone microenvironment around the drug-loaded titanium implant Significant changes will have a certain impact on the process of drug release, diffusion, and absorption into the blood【DOI: 10.1016/j.msec.2017.08.056】. Conversely, the above process causes the concentration change around the tita-nium implant to also affect the healing effect of the bone. The relationship between the release of drugs under the control of implants and the process of osseointegration is ex-tremely complicated in time and space. The purpose is to balance the drug release and osseointegration effect of the titanium implant. The current amount of research is rela-tively small, and it is still a issues to be explored in the future.

Comments and suggestions ⑤:

1,Although the authors make an adequate analysis of what is addressed in the review, this resembles more the comments of the authors that a discussion supported or contrasted with the existing bibliography.

2,In general, in addition to an excessively broad review of the local drug delivery systems literature on titanium implants, it does not provide news, also using 50% of bibliographic references older than 5 years

Response1: Thank you for your comment. We have modified the Discussion, added supporting references and other content including more comprehensive processing technology of titanium implants and prospected direction (magnetron sputtering technology, plasma spraying technology, etc.); more comprehensive clinical problems; part of the molecular mechanism of osseointegration (histological level, Vascularization related issues) and many other issues about orthopedic clinic. the modified Discussion now is: 

Disccusion

From the clinical perspective of orthopedics, the development of local drug delivery systems on titanium implants has given orthopedic treatment a new, feasible and effective direction. This inevitably needs to balance the following four challenges: the pharmacological properties of loaded drugs, the molecular biological mechanism of bone cells and surrounding tissues, the processing technology of titanium implants and the correct choice of clinical application.

Loaded Drugs and Their Pharmacological Properties

In the clinical experience of orthopedics, the drugs frequently used with titanium implant surgery are mainly divided into several categories, analgesic and anti-inflammatory drugs (especially NSAIDs, acetaminophen and opioids are the most common), and some anti-bone tumors Drugs (doxorubicin, cisplatin and curcumin), antibiotics (vancomycin and clindamycin), anabolic drugs (calcitriol, teriparatide and fluvastatin), anti-catabolic drugs (alendronate sodium) , Zoledronate and raloxifene), dual anabolic and anti-catabolic drugs (simvastatin and strontium ranelate). Due to the pharmacological properties of these drugs, they are usually selected as systemic drugs or drugs loaded with titanium implants to achieve the purpose of personalized treatment. However, many reports have confirmed that the properties of these drugs sometimes affect titanium. The osseointegration effect of the implant interface, which deserves attention. NASAIDs control inflammation and pain by regulating the COX-2 pathway in arachidonic acid, thereby reducing the synthesis of prostaglandins, but Chikazu et al. and Zhao et al. reported that the lack of COX-2 can cause small adhesion and aggregation of osteoblasts around the mouse titanium implant is reduced, and the effect of osseointegration is weakened 112,113, but in the report of Yang et al., it is further believed that the controlled dose of NSAIDs has an effect. The key to the osseointegration effects 114. In response to the above problems, the advantages of acetaminophen are shown. It has a low effect on the COX-1 and COX-2 pathways. Although it is weak in inhibiting the synthesis of prostaglandins, it still exists. It has a certain anti-inflammatory and analgesic effect. It may be a correct choice for patients with non-severe pain and need to enhance the rate of osseointegration, which may be a hint for the subsequent choice of anti-inflammatory drugs for titanium implants effect. Teriparatide is a kind of anabolic drugs, which acts on the PTH receptor of osteoblasts, thereby affecting the level of parathyroid hormone to regulate bone production. However, when teriparatide is applied in large doses, it can cause osteoclasts. Extensive activation, and will cause the expression levels of specific transcription factors, osteocalcin, bone salivary protein and type I collagen in osteoblasts to be down-regulated to varying degrees, tending to bone catabolism. These conclusions are corroborated in the reports of Almagro et al. and Tao et al 115,116. Therefore, after the titanium implant is loaded with teriparatide, it is necessary to strictly control the local concentration of the drug around the implant. Zoledronate is an excellent anti-catabolic drug, which has a high affinity for mineralized bone. It targets farnesyl pyrophosphate synthase in osteoclasts, and then inhibits the activity of osteoclasts or directly Induces the apoptosis of osteoclasts to inhibit the bone resorption of osteoclasts 117, so scholars usually choose them as loading drugs for titanium implants. However, according to reports by Basudan et al. and Cardemil et al., zoledronate has different osseointegration effects on bone tissues in different parts of the human body. For example, the mandible and tibia and fibula have negative effects. Therefore, the application of zoledronic acid in different implant sites needs to be individually adjusted. The pharmacological properties of the drug itself and titanium implants may interfere with each other. Nowadays, there is relatively little exploration in this field. For example, whether anti-tumor drugs, opioids, etc. will affect the properties of titanium implants, and then Affect the osseointegration effect? It is necessary to be more cautious in the choice of implant loading drugs, and more future experiments are needed. In addition, scholars have paid more and more attention to melatonin in the treatment of bone tumors in recent years, but the use of melatonin in the local drug delivery of titanium implants is rare 118.

Molecular Biological Mechanism of Bone Cells and surrounding tissues

From the perspective of molecular biology of osteoblasts, osteoclasts, fibroblasts and mesenchymal stem cells, the mechanisms of bone aggregation, osteoinduction, and osseointegration on titanium implants have been deeply studied119-122. However, in recent years, there have been more and more explorations on the role of integrins on the surface of bone cells. Olivares-Navarrete et al. reported that they prepared titanium implants attached with a graphite carbon coating and studied the important role of integrins in the process of bone cells maturation and osseointegration on the surface. Integrin is a transmembrane heterodimer, and each subunit has its unique role. It is because of its existence that osteoblasts and mesenchymal stem cells are connected with extracellular proteins, including actin, Laminin, etc., so it regulates the subsequent important processes of bone cell attachment, aggregation, and migration. The experimental results of this study also indicate that the integrin α subunit plays a major role in surface chemical recognition. And within a certain range, the roughness of the titanium implants can enhance the effect of bone differentiation and osseointegration123. The exploration from the perspective of molecular biology also provides instructions for the selection and application of implants. For example, due to the surface chemical recognition of integrin α subunits, it is possible to choose to loading drugs or other agents to target the integrin α subunit so that affect the maturation of osteoblasts. It is also possible to reasonably increase the surface roughness of the titanium implant to enhance the osseointegration effect. In order to play a prompting role, the molecular biological mechanisms of bone growth, osteoinduction, and osseointegration on the surface of titanium implants still need to be further explored. In addition, as for the related issues of vascularization at the histological level (relevant mechanisms have been elaborated in 4.1), there are few researches and few related commercial drugs, which is expected to become a key research direction in the future.

Processing Technology of Titanium Implants

The physical modification of titanium implants, surface modification and other processing technologies play an important role in its own antibacterial properties and the ability to promote osseointegration. It has summarized that the predecessors prepared the TiO2 nanotube structure to carry drugs through different metal processing techniques such as the micro/nano-level modification technology of the titanium surface, and adjusted the diameter, vertical depth and tube wall thickness of the titanium nanotubes to control the drug Load or its release, preparation of ceramic thin film layers by electrochemical deposition to store drugs And by preparing a coating structure on the surface of the implant, using the ability of the coating structure to load the drug to load and control the release of the drug. Generally speaking, the common challenges focus on how to solve the stability of drug loading and long-term slow drug release, and whether the physical properties of titanium implants, such as elastic modulus, corrosion resistance, and fatigue resistance, can meet the needs of patients. At the same time, under different demand conditions, the requirements for the pore size, porosity and internal spatial structure of titanium implants (titanium scaffolds) are also different. Under this complex structure, the control of drug loading and release has become new challenges. All these require advances in materials engineering and metal processing technology, although there have been many breakthroughs in anodic oxidation technology, micro-arc oxidation technology, covalent grafting technology and Layer-by-Layer technology. However, the exploration of metal engineering scholars is still needed. Each part of the review describes in detail the difficulties faced by each achievement and the solutions of existing metal processing technologies, so it won't repeat them here. The comparison of various titanium implant processing techniques is also underway, for example, some studies reported that the efficiency of surface modification of Ti-based implants is not as effective as surface roughening124,125.

But it is worth mentioning that, after implantation, the interaction between the titanium implant and the human body will produce various complex biological, physiological and chemical reactions. In addition to the antibacterial, anti-tumor and osseointegration capabilities of titanium implants that are the most important in the field of orthopedics, the better biocompatibility of titanium implants is also an indispensable character. RF magnetron sputtering technology, Plasma spraying technology and other technology also play an important role in the preparation of titanium implants,in some reports, achievements have also been made to enhance the biocompatibility of titanium implants 126-129. In summary, in orthopedics clinics, in order to meet the needs of orthopedics, the processed titanium-based implants should have physical properties including elastic modulus, fatigue resistance, etc.; biological properties including antibacterial properties, osseointegration capabilities, and compatibility, etc.; other necessary characteristics including drug carrying capacity and corrosion resistance. The processing technology of titanium implants should balance these characteristics of the implants, so that the necessary indicators can achieve satisfactory results. However, this is also the focus and difficulty in this field, and further research and exploration are still needed.

However, from another perspective, some innovative attempts are gradually emerging. Some researchers are dedicated to processing variety of Ti-based alloys. They use various techniques mentioned above to obtain titanium substrates loaded with different chemical compositions or organics instead of simply physically modified titanium 130, This seems to open up a whole new research direction. However, considering the clinical perspective of orthopedics, how to ensure the biological safety, versatility and effectiveness of these chemical components and loaded drugs may be the focus.

Clinical Application of Titanium Implants

In orthopedic clinics, it is also an art to provide personalized titanium implants for patients with different needs to achieve the purpose of treatment. The patients who need implant surgery are usually patients who have undergone massive bone resection of bone tumors, patients with severe osteomyelitis,  patients with bone defects caused by trauma and so on. As for patients with bone tumors, titanium implants carrying anti-tumor drugs (doxorubicin, cisplatin, curcumin, etc.) are usually used. Titanium implant scaffolds are also required for better osseointegration efficiency because of their anti-tumor effects. Long-term effective local delivery of drugs and rapid osseointegration achieve better results. Combination of chemotherapy drugs and surgery has become one of the important methods of bone tumor treatment. As for patients with severe osteomyelitis, because the tissue surrounding the titanium implant is already in a state of bacterial infection, delayed or nonunion of the bone is very prone to occur. Orthopedics usually choose to carry multiple antibiotics (vancomycin, gentamicin, etc.) Coating-loaded titanium implants, combined with antibiotic bone cement and other materials for treatment, use the continuous and slow release of antibiotics around the titanium implants to keep the local antibiotic level at a high level, which is beneficial to patients with osteomyelitis for recover. As for patients with large bone defects caused by trauma, if the distance between the fracture ends exceeds 1-2cm, they cannot heal by themselves. Orthopedics usually choose titanium implants loaded with  bone anabolic drugs, anti-catabolic drugs or growth factors(rhBMP2, zoledronate, simvastatin, calcitriol, etc.). At the same time, titanium implant scaffolds are required to have suitable physical properties (larger porosity, higher roughness, etc.) to assist the slow local delivery of bone growth drugs to better promote bone growth and osseointegration.

It cannot be ignored that it is very meaningful to explore the effect of local loading of anti-tumor drugs combined with systemic chemotherapeutics on titanium implants at different stages of the tumor. However, there are very few relevant studies at present. We believe that the reason for this issue is that the number of patients is small (the number of patients with bone tumors is small), the severity of tumors is different, the sensitivity of individual drugs is different (whether Homo sapiens or experimental animals), and the constraints of relevant ethical laws, etc. It is very difficult to organize a large sample of research and it is necessary to overcome the various problems mentioned above. Therefore, the anti-tumor synergistic effect of systemic chemotherapy and local drug delivery system has no accepted conclusion, which requires a large number of long-term studies.

With the development of pharmacology and pharmacokinetics, the drug itself is constantly innovating to adapt to various implant environments. At the same time, with the development of metal materials science, the processing technology of titanium implants is gradually innovating, and various devices that conform to human biomechanics and are more ingenious to store and slow-release drugs have been prepared. This allows the drug delivery system to have a longer action time and a more stable drug output. In the future, more research should focus on the "holistic nature" of the synergy between titanium implants and their drug-loading systems, to achieve the antibacterial properties of the implants, the ability to promote osseointegration, the balance of physical properties and other personalized requirements, and comprehensively Solve the various required properties of implants. On the basis of today, combined with the latest developments in pharmacology and metal materials science, we can synergistically solve more problems in the field of orthopedics.

Response2: Thank you for your comment. According to your suggestion, we have already addressed relative issues about Reference.

Thank you again for these instructive suggestions.

Round 2

Reviewer 1 Report

can now be accepted.

Author Response

Thank you for your comments.

Reviewer 2 Report

The manuscript corrected is good.

One remark. The short section "Conclusions" should be added.

Author Response

Comments: The manuscript corrected is good. One remark. The short section "Conclusions" should be added.

Response: Thank you very much for your comments. We have revised the manuscript according to your suggestions. All the changes of the manuscript have been marked red and the response to the questions arisen from the review comments has been highlighted in Blue.

Response: Thank you for your comment. We have added the conclusion and summarized the full text in it. the added information is (On Page 22 Line915-928):

Conclusion

With the development of pharmacology and pharmacokinetics, the drug is constantly innovating to adapt to various implant environments. At the same time, with the development of metal materials technology, the processing technology of titanium implants is gradually innovating, and various devices that conform to human biomechanics and are more ingenious to store and slow-release drugs have been prepared. Which allows the drug delivery system to have a longer action period and a more stable drug output. In the future, more research should focus on the "holistic nature" of the synergy between titanium implants and their drug-loading systems, to achieve the antibacterial properties of the implants, the ability to promote osseointegration, the balance of physical properties and other personalized requirements, and comprehensively solve the various required properties of implants. From the perspective of the orthopedics field so far, combined with the latest developments in pharmacology and metal materials science, we can synergistically solve more problems in the field of orthopedics.

Reviewer 3 Report

The review is correct

Author Response

Thank you for your comment.